# Domain Adaptation for Cold-Start Users in Sequential Recommendation

**Lu Wang**  *wang_lu@a-star.edu.sg*
*Institute for Infocomm Research ($I^2R$)*
*A\*STAR, Singapore*

**Wenyu Zhang**  *wz258@cornell.edu*
*Institute for Infocomm Research ($I^2R$)*
*A\*STAR, Singapore*

**Chengke Wang**  *wangchengke@u.nus.edu*
*School of Computing*
*National University of Singapore*

**Guimei Liu**  *liug@a-star.edu.sg*
*Institute for Infocomm Research ($I^2R$)*
*A\*STAR, Singapore*

**Ye Luo**  *yeluo@tongji.edu.cn*
*School of Computer Science and Technology*
*Tongji University, China*

**Reviewed on OpenReview:** *https://openreview.net/forum?id=jrVmW6LfMa*

## Abstract

Sequential recommendation tracks users' preferences over time based on users' historical activities and makes prediction on their next most probable action. However, this approach faces limitations when dealing with cold-start users who possess minimal interaction data, leading to difficulty in learning their preferences. To address this challenge, by taking regular users with longer interaction histories and cold-start users as two domains, this paper introduces domain adaptation techniques to narrow the performance gap caused by knowledge shifts in domains. We propose a dual-transformer framework with separate models for long (source) and short (target) sequences, collaboratively trained with shared item embeddings. To enable effective knowledge transfer, we introduce an emulated target domain by sampling short sequences from the source, and apply contrastive learning to align their contextual representations. To further improve adaptation under compound knowledge shifts, we reduce item popularity bias and incorporate user similarity into the contrastive loss. Experiments on five public datasets show consistent improvements over strong baselines, demonstrating the robustness of our approach under both length shifts and compound shifts involving item distribution changes.

## 1 Introduction

Recommender systems help users navigate overwhelming item choices by modeling preferences from historical interactions. Traditional methods such as collaborative filtering Wang et al. (2006); Hu et al. (2008) and generative models Liang et al. (2018) typically assume past interactions are equally important. In contrast, modern sequential recommendation methods model users' evolving interests using sequence models, particularly transformers Wang et al. (2019); Fang et al. (2020); Hidasi et al. (2016); Donkers et al. (2017); Hidasi &

Karatzoglou (2018); Kang & McAuley (2018); Li et al. (2020); Sun et al. (2019); Xie et al. (2022). A major challenge arises in handling cold-start users, those with sparse history, compared to regular users who have rich behavioral data. Standard models, trained on all users, tend to be dominated by regular users, leading to biased learning and poor generalization to cold-start users.

We tackle the cold-start sequential recommendation problem through the lens of domain adaptation (DA), framing regular users as the source domain and cold-start users as the target domain. This perspective highlights two key types of domain shifts: length shift, where cold-start users have much shorter interaction histories, and complex shift, where item distributions also differ due to temporal item evolution, in addition to sequence length differences.

To address these challenges, we propose a contrastive DA framework built on dual transformer networks: one specialized for long sequences in the source domain, and the other for short sequences in the target domain. To bridge the gap between domains, we introduce an emulated target domain by sampling short subsequences from the source. We then apply contrastive learning to align the contextual representations of these sampled sequences with their original long counterparts, enabling effective and controllable knowledge transfer. To improve adaptation under complex shifts, we further reduce item popularity bias in user representations and incorporate user similarity into the contrastive objective. These enhancements progressively increase robustness to distributional shifts and promote more personalized, domain-invariant knowledge alignment. Our contributions are as follows:

- We reformulate cold-start sequential recommendation as a domain adaptation task, identifying length shift and compound shift as the two key knowledge transfer challenges.
- We propose a simulated-target training bridge that constructs short subsequences of source users as labelled cold-start examples, providing direct supervision in the length regime absent from standard training.
- We introduce a source-side, similarity-weighted contrastive objective that imposes alignment between full-length and short-length representations of the same user, avoiding the instability of symmetric cross-domain similarity when the target side has few interactions.
- We instantiate these mechanisms in three progressive model variants (DACSR, DACSR+, DACSR++) that incrementally incorporate popularity-bias reduction and similarity-weighted alignment for increasingly fine-grained knowledge transfer.
- Experiments on public datasets demonstrate that the proposed models consistently outperform baselines under both length and compound shift settings, whereas standard transfer learning degrades significantly under compound shift.

## 2 Related Work

We organize prior work into four families and make the applicable scope explicit. Our setting uses only user-item interaction sequences: no user/item features, no knowledge graphs, and no user- or item-overlap assumption between the source and target populations. Under this scope, methods that rely on side information, shared item spaces, or a shared-user bridge are not directly comparable; we still discuss them because they situate our contribution.

### 2.1 Sequential Recommendation

Sequential recommenders model temporal behavior for next-item prediction. Early approaches are based on Markov Chains and hybrids such as FPMC and MFMC Shani et al. (2002); Rendle et al. (2010); He & McAuley (2016). Deep models capture richer dependencies: CNN-based Caser and SimpleCNN Tang & Wang (2018); Yuan et al. (2019a), RNN-based GRU variants Hidasi et al. (2016); Donkers et al. (2017); Hidasi & Karatzoglou (2018), and transformer-based SASRec, BERT4Rec, and TiSASRec Kang & McAuley (2018); Sun et al. (2019); Li et al. (2020), which have become dominant. Recent architectural variants of this family replace self-attention with lighter alternatives of linear attention in LinRec Liu et al. (2023) and a selective-gated Mamba / state-space backbone in SIGMA Liu et al. (2024), but they remain single-tower sequential encoders trained on pooled interaction data without an explicit transfer mechanism. When trained

on the union of source and target interactions, this family acts as an *implicit* knowledge-transfer baseline: any benefit from source users is only through pooled-data exposure.

Contrastive learning (CL) and self-supervision further strengthen sequential encoders. CL4SRec Xie et al. (2022) applies crop/mask/reorder augmentations with a contrastive objective; S3-Rec Zhou et al. (2020) aligns attribute signals with sequences; DuoRec Qiu et al. (2022) improves representation uniformity by pairing sequences with the same terminal item; ICLRec Chen et al. (2022) enforces intent-prototype consistency. These methods improve robustness of a single sequential encoder but treat all users uniformly and do not impose any alignment between a long-history *source* regime and a short-history *target* regime.

## 2.2 Cold-Start Recommendation

Cold-start methods fall into two broad categories: those that incorporate additional information and those that improve model generalization including cross-domain methods.

**Incorporating Additional Information.** A common approach leverages side information such as user profiles or item content in collaborative filtering Gantner et al. (2010); Li et al. (2019) and deep models Barkan et al. (2019); Zhu et al. (2020); Wei et al. (2021), aligning auxiliary features with collaborative embeddings. Contrastive learning has been used for feature alignment across datasets Wei et al. (2021), and domain adaptation Zhu et al. (2021); Khan et al. (2017) and those with side information Chen et al. (2023a); Zhao et al. (2020); Yuan et al. (2019b); Chen et al. (2021) has been applied to transfer from related tasks. Under our interaction-only scope, methods in this family are out of scope as general-purpose baselines.

**Improving Model Generalization.** A second strategy is to improve robustness to unseen users or items. Du et al. (2020) addresses train/test embedding discrepancies, and DropoutNet Volkovs et al. (2017) randomly masks collaborative signals during training. Meta-learning Hospedales et al. (2022) methods such as MeLU Lee et al. (2019), Vartak et al. (2017); Wei et al. (2020) use MAML Finn et al. (2017) to adapt quickly to new users; most variants require user/item attributes. MetaTL Wang et al. (2021) is the interaction-only sequential cold-start representative of this family and is therefore directly comparable. Mecos Zheng et al. (2021) uses a prototypical network with a transformer backbone for cold-start items.

**Cross-Domain Sequential Recommendation.** Cross-domain sequential recommendation (CDSR)Zhu et al. (2021) traditionally bridges two domains via shared/overlapping users. EMCDR Man et al. (2017), Chen et al. Chen et al. (2023b) and CDRIB Cao et al. (2022) learn a mapping between user representations across domains and therefore require a substantial shared-user base. More recently, AMID Xu et al. (2024) relaxes this to an open-world assumption in which only partial user overlap is available: it constructs interest groups via inner- and inter-domain compositions that accommodate both overlapping and non-overlapping users, and debiases the observed cross-domain signal. CDSR methods that further require item-side knowledge graphs or content features, e.g., HeroGraph Mukande (2022), MIFN Ma et al. (2022) and BiTGCF Liu et al. (2020) are out of scope in our benchmark. Our setting is stricter than AMID's: source users and target cold-start users are *disjoint* populations. AMID is nevertheless the closest available CDSR baseline because its open-world design does not explicitly require a shared-user bridge.

## 2.3 Positioning of DACSR

Within the applicable scope, the directly comparable baselines fall into different groups: (i) implicit combined training via recent sequential encoders (SASRec, LinRec, SIGMA), (ii) explicit source-assisted transfer learning, (iii) interaction-only meta-learning (MetaTL Wang et al. (2021)), (iv) contrastive self-supervised sequential methods (CL4SRec), and (v) open-world CDSR (AMID).

DACSR differs from all in a structural way. (a) Unlike (i) and (ii), it does not treat the source only as extra training data or as a pretraining init; it constructs a *simulated target domain*, i.e., short subsequences of source users, to provide labelled examples in the cold-start length regime during training. (b) Unlike (iii), it does not rely on episode-level task construction but imposes a continuous similarity-weighted alignment objective over the source side. (c) Unlike (iv), its contrastive signal is not within-sequence augmentation robustness

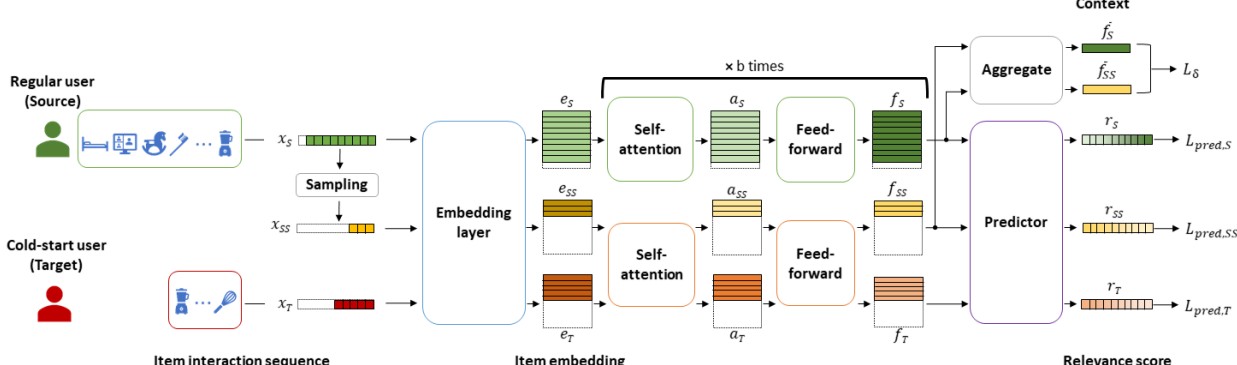

Figure 1: The DACSR framework consists of four main groups of components to predict the relevance of items at the next interaction: A subsampling operator ('Sampling') on the source (S domain) to generate an emulated cold-start target or sampled source (SS domain), embedding layer and predictor layer shared by all domains, $b$ transformer blocks for the source and another $b$ blocks shared by the emulated cold-start target and the target (T domain), and a measure of divergence $\delta$ between the contexts of the original source and the emulated target.

but cross-view alignment between a user's full-length and short-length representations. (d) Unlike (v), it does not use similarity as a symmetric selection gate across domains, which is unstable when the target side has few interactions, but as a *unidirectional, source-side* optimization signal that regularizes the representation space. The contributions are therefore the three concrete mechanisms that realize that framing: a simulated-target training bridge, a source-side similarity-weighted contrastive objective, and explicit length-invariant same-user alignment.

## 3 Methodology

We address the cold-start sequential recommendation problem, where users with limited interaction histories (cold-start users) coexist with users who have rich behavioral data (regular users). In our setting, sequences from regular users define the source domain, while those from cold-start users define the target domain. This domain shift is characterized by differences in sequence lengths and potentially item distributions over time.

### 3.1 Knowledge Shift and Transfer

**Length Shift.** We propose a dual-network framework with separate source and target encoders. The two encoders share item embeddings but maintain domain-specific parameters, allowing each encoder to specialize in its own domain while still benefiting from joint learning. To transfer knowledge from long source sequences to short target sequences, we construct an emulated target domain by randomly sub-sampling source sequences with a controllable keep probability. This produces short source-derived sequences whose lengths approximate those of real cold-start users. The emulated target domain therefore bridges the length gap between the source and target domains and improves generalization to truncated interaction histories.

**Compound Shift: Length and Item-distribution Changes.** In more realistic settings, the source and target domains may differ not only in sequence length but also in item distribution. To handle this compound shift, we further introduce domain-invariant transfer mechanisms. First, we compute item popularity separately in each domain and use popularity-weighted user representations for domain alignment. This reduces sensitivity to uneven item exposure and encourages the model to focus on transferable preference structure rather than domain-specific popularity bias. Second, we introduce user similarity-aware contrastive alignment to preserve source-domain preference relations in the short-sequence regime.

The emulated target domain serves two key purposes. First, it enables length-invariant user alignment: a user's full source sequence and its simulated short sequence are encouraged to produce consistent represen-

tations. This trains the target encoder to be robust to the length gap between long-history source users and cold-start users. Unlike generic sequence augmentations in conventional contrastive learning, this alignment explicitly transfers per-user long-sequence knowledge to the target encoder used for cold-start recommendation. Second, it enables cross-user similarity transfer. Users that are similar in the source domain, estimated from their long histories, are encouraged to remain close in the simulated short-sequence space. This preserves preference structure under sparsity and improves generalization to real cold-start users at inference time.

### 3.2 Dual-network Architecture

Our DACSR (Domain Adaptation for Cold-Start Recommendation) is a dual-transformer framework, which explicitly designed to transfer knowledge from regular to cold-start users. As illustrated in Figure 1, DACSR employs two domain-specific networks to model the distinct data distributions of the source and target domains along with an emulated target domain, formed by randomly sampling from source sequences with a controllable probability. The dual network of source and target share item embeddings with separate parameters, allowing each to specialize in its respective domain while training jointly. Both source and target adopt transformer architectures Kang & McAuley (2018); Zhang et al. (2019) due to their strength in modeling long-range dependencies.

Let $U$ and $V$ denote the sets of users and items, respectively. A user $i \in U$ has a chronologically ordered interaction sequence $x_i = (v_1, v_2, \ldots, v_L)$, where $v_l \in V$ is a one-hot vector denoting the item interacted at time $l$. The goal of sequential recommendation is to predict the next item a user is likely to interact with, using only historical sequences and without side information. Each item $v \in V$ is embedded into a $d$-dimensional vector using an embedding matrix $\mathbf{M} \in \mathbb{R}^{|V| \times d}$. Positional information is added using a trainable matrix $\mathbf{P} \in \mathbb{R}^{L \times d}$, resulting in the final sequence embedding:

$$\mathbf{E} = [\mathbf{m}_{v_1} + \mathbf{p}_1, \mathbf{m}_{v_2} + \mathbf{p}_2, \ldots, \mathbf{m}_{v_L} + \mathbf{p}_L], \tag{1}$$

where $\mathbf{m}_{v_l} \in \mathbf{M}$ and $\mathbf{p}_l \in \mathbf{P}$. Sequences are padded or truncated to a fixed length $L$.

The embedding $\mathbf{E}$ is passed through $b$ stacked transformer blocks. Each block consists of a self-attention layer followed by a feed-forward layer. The attention is computed via:

$$\mathbf{S} = \text{Attention}(\mathbf{Q}, \mathbf{K}, \mathbf{V}) = \text{softmax}\left(\frac{\mathbf{Q}\mathbf{K}^T}{\sqrt{d}}\right)\mathbf{V}, \tag{2}$$

where $\mathbf{Q} = \mathbf{E}\mathbf{W}^Q$, $\mathbf{K} = \mathbf{E}\mathbf{W}^K$, $\mathbf{V} = \mathbf{E}\mathbf{W}^V$, and $\mathbf{W}^Q, \mathbf{W}^K, \mathbf{W}^V \in \mathbb{R}^{d \times d}$ are learnable parameters. To preserve the autoregressive property of sequential recommendation, we apply a causal attention mask to prevent a position from attending to future tokens.

Each output $\mathbf{s}_l \in \mathbf{S}$ is passed through a feed-forward network:

$$\boldsymbol{f}_l = \text{ReLU}(\mathbf{s}_l \mathbf{W}_1 + \mathbf{b}_1)\mathbf{W}_2 + \mathbf{b}_2, \tag{3}$$

where $\mathbf{W}_1, \mathbf{W}_2 \in \mathbb{R}^{d \times d}$ and $\mathbf{b}_1, \mathbf{b}_2 \in \mathbb{R}^d$ are shared across all positions. The output of the final transformer block at position $l$ is used to compute the preference score for the next item using a dot product with the item embedding:

$$\boldsymbol{r}_{l,v} = \boldsymbol{f}_l \cdot \mathbf{m}_v, \tag{4}$$

where $\mathbf{m}_v \in \mathbf{M}$. We reuse the item embedding matrix in the output layer for efficiency and to reduce overfitting, following Kang & McAuley (2018). This dual-network structure, with shared embeddings and a common predictor, allows the model to capture domain-specific patterns while enabling shared learning across both domains. The following sections will describe how we align these domains using contrastive learning and explicitly regulate the transfer process.

### 3.3 Emulated Cold-Start Target Domain

To explicitly control knowledge transfer from long to short sequences, we introduce an emulated target domain by sampling short sequences from long user histories in the source domain. These sampled sequences,

representing cold-start behaviors, are input into the target model, while their original long counterparts are processed by the source model. By aligning the contextual representations of similar sampled and long sequences, we aim to regulate the amount of knowledge transferred.

As illustrated in Figure 1, we apply a sampling operator to each long sequence from regular users using a uniform probability $p$, a hyperparameter that controls the average length of the emulated short sequence. The sampled interactions are ordered chronologically to form the sampled source (SS) sequence. For each user, the original long sequence (S) is processed by the source model, while the sampled short sequence (SS) and the actual target sequence (T) are processed by the target model. The SS domain serves as an intermediate bridge between the source (S) and target (T) domains.

We currently adopt a uniform random subsampling strategy for $p$ to construct emulated target sequences. An alternative is to use target-aware or temporally stratified sampling, where source users are grouped by recency and sampled with learned probabilities following concept-drift-aware data selection Kim et al. (2024). We explore this extension in supplementary experiments and observe modest gains in certain settings; full details are provided in the Appendix I.

### 3.4 Domain Alignment

Let $\mathbf{F}^D = [\boldsymbol{f}_1, \ldots, \boldsymbol{f}_L]$ denote the sequence of contextual outputs from the final transformer layer in domain $D \in \{S, SS\}$. To minimize the representation gap between long and short sequences for similar users, we align $\mathbf{F}^S$ and $\mathbf{F}^{SS}$ via contrastive learning, operating on user-level representation vectors $\bar{\mathbf{F}}^D$ aggregated from the sequence outputs.

**User-level Aggregation.** We consider two aggregation strategies. The first is a simple average

$$\bar{\mathbf{F}}^D = \mathbf{Avg}(\mathbf{F}^D). \tag{5}$$

The second is a popularity-adjusted average

$$\bar{\mathbf{F}}^D = \frac{1}{L} \sum_i \boldsymbol{f}_i w_i, \tag{6}$$

where the weight $w_i$ is a bounded inverse of item popularity:

$$w(i) = \begin{cases} 0.1/\mathrm{P}_i & \text{if } \mathrm{P}_i \geq 0.1, \\ 1 & \text{if } \mathrm{P}_i < 0.1, \end{cases}$$

with $\mathrm{P}_i$ the normalized popularity count of item $i$ across all users. All weights lie in $[0.1, 1]$, giving a maximum amplification ratio of $10\times$ between rare and popular items. This acts as a bounded rebalancing mechanism that increases the influence of less popular items without suppressing frequent ones, preserving diversity rather than collapsing toward rare-item embeddings.

**Contrastive Losses.** Two contrastive losses are employed. The *traditional in-batch contrastive loss* defines positive pairs as long–short sequence pairs from the same user, and negative pairs as sequences from different users within the batch. The NT-Xent loss Chen et al. (2020) encourages alignment of positive pairs while separating negatives:

$$L_\delta^{in} = -\sum_i \sum_j \log \frac{\exp(sim(\bar{\mathbf{F}}_i, \bar{\mathbf{F}}_j)/\tau)}{\sum_{k=1}^{2N} 1_{k \neq i} \exp(sim(\bar{\mathbf{F}}_i, \bar{\mathbf{F}}_k)/\tau)}, \tag{7}$$

where $\bar{\mathbf{F}} = [\bar{\mathbf{F}}^S, \bar{\mathbf{F}}^{SS}]$ is the concatenated batch of size $2N \times d$, $(i, j)$ index positive pairs, $sim$ denotes cosine similarity with temperature $\tau$, and $1_{k \neq i}$ is the standard indicator function.

The *similarity-weighted contrastive loss* instead treats all user pairs as positives, weighted by precomputed similarity scores adapted from He et al. (2022), so that closer users are pulled together more strongly for finer-grained alignment:

$$L_\delta^s = -\sum_i \sum_{j \neq i} S_{ij} \log \frac{\exp\big(sim(\bar{\mathbf{F}}_i^S, \bar{\mathbf{F}}_j^{SS})/\tau\big)}{\sum_{k=1}^{N} 1_{k \neq i} \exp\big(sim(\bar{\mathbf{F}}_i^S, \bar{\mathbf{F}}_k^{SS})/\tau\big)}, \tag{8}$$

where $S_{ij}$ is the pairwise similarity score computed over full source sequences:

$$S_{ij} = \frac{\exp\big(sim(\bar{\mathbf{F}}_i^S, \bar{\mathbf{F}}_j^S)/\tau\big)}{\sum_{k=1}^{N} \exp\big(sim(\bar{\mathbf{F}}_i^S, \bar{\mathbf{F}}_k^S)/\tau\big)}.$$

Similarity scores are computed on full source sequences rather than short sampled ones, as the latter tend to introduce high variance and yield unstable estimates. Moreover, since these scores are learned by the model, they are unreliable early in training. We therefore employ an annealed interpolation between $L_\delta^{in}$ and $L_\delta^s$ until the similarity estimates stabilize; the detailed training strategy is described in Appendix A.

### 3.5 Final Loss Function

We train the model with the set of users' interaction sequences $x_i = (v_1, v_2, \ldots, v_L)$. An input sequence from a user is $(v_1, v_2, \ldots, v_{L-1})$ without the last interaction, and the target sequence for prediction, $y_i = (v_2, v_3, \ldots, v_L)$, is the right shifted version of the input sequence. To assess the predictions of each domain, we employ the cross-entropy (CE) loss, which is defined as:

$$L_{pred,D} = \frac{-1}{N(L-1)} \sum_{i=1}^{N} \sum_{l=1}^{L-1} \sum_{v \in V} y_{i,l,v} \log \frac{\exp(r_{i,l,v})}{\sum_{j=1}^{|V|} \exp(r_{i,l,j})}, \tag{9}$$

where $D \in \{S, SS, T\}$, $N$ is the batch size with $i \in \{1, \ldots, N\}$, $L$ is the sequence length with $l \in \{1, \ldots, L-1\}$, $V$ is the set of items from interactions, $y_{i,l,v}$ is the indicator and $r_{i,l,v}$ is the relevance score of item $v$ for user $i$ at time $l$ given in Eq.(4).

The overall loss function to be minimized is then

$$L = L_{pred,S} + \beta_{SS} L_{pred,SS} + \beta_T L_{pred,T} + \lambda L_\delta, \tag{10}$$

where $\beta_{SS}$, $\beta_T$, $\lambda$ are hyperparameters, $L_{pred,S}$, $L_{pred,SS}$ and $L_{pred,T}$ are the CE loss of source, emulated target and target, respectively, and $L_\delta$ is the contrastive loss in either Eq.(7) or Eq.(8). Discussions on the hyperparameters and their impacts are given in Appendix B.

### 3.6 Model Variants

Depending on whether popularity bias reduction is applied (via Eq.(5) vs. Eq.(6)) and whether user similarity is incorporated in the contrastive loss (Eq.(7) vs. Eq.(8)), we implement three model variants that progressively improve robustness to different knowledge shifts:

- **DACSR**: Uses averaged user representation in Eq.(5) and traditional in-batch contrastive loss in Eq.(7).

- **DACSR+**: Applies popularity-weighted user representation in Eq.(6) to reduce item bias, while retaining the traditional contrastive loss in Eq.(7).

- **DACSR++**: Combines popularity-reduced user representation in Eq.(6) with similarity-weighted contrastive loss in Eq.(8) for more fine-grained and personalized knowledge transfer.

## 4 Experimental Results

### 4.1 Datasets and Splits for Cold-start

We evaluate our models on five public Amazon product review datasets McAuley et al. (2015): Toys, Electronics, Beauty, Sports, and Kindle, treating reviews or ratings as implicit feedback and using timestamps to determine interaction sequences. These datasets were selected for their public availability and their widespread use as benchmark collections in recommender-system research Kang & McAuley (2018); Wang

Table 1: Jaccard indexes of the processed datasets. Lower values indicate greater item distribution shifts. Compared with the random split, the time-based split introduces a more severe distribution shift.

| Split | Setting | Electr | Kindle | Beauty | Toy | Sports |
|---|---|---|---|---|---|---|
| *Random* | - | 0.6949 | 0.3986 | 0.5083 | 0.3986 | 0.6260 |
| *Time-based* | Filtering | 0.3423 | 0.2195 | 0.1905 | 0.1696 | 0.3513 |
| | No filtering | – | 0.1976 | 0.1299 | 0.1299 | 0.3245 |

et al. (2021). Following Wang et al. (2021), we preprocess each dataset to ensure sufficient sequence lengths and data quality. Refer to Appendix C for detailed dataset statistics.

To simulate different domain shift scenarios, we apply two splitting strategies to divide users into cold-start (target) and regular (source) domains:

- Random Split (Length Shift): Users with fewer than 10 interactions are filtered out. We randomly select 20% of users as the target set (cold-start) and the rest as source (regular). Items not shared between domains are removed to minimize item shift. The maximum target sequence length is set by a parameter *max_len*, controlling the degree of length shift. The random split should be interpreted as a controlled target-length reduction (i.e., length-shift) experiment, rather than a realistic cold-start scenario. The short sequence length primarily reflects data sparsity instead of a true distribution shift, although a mild shift may still be introduced in this setting.

- Time-Based Split (Compound Shift): Users who interacted before 1/1/2014 are source users; others are targets. Items with fewer than 10 reviews are filtered, and user sequences shorter than 10 (source) or 4 (target) are discarded. This approach reflects real-world cold-start settings, where newer users face both shorter sequences and changing item distributions.

We define two item-distribution variants under the time-based split: (1) Filtering, where items in the source that are not present in target are filtered (considering shared item set), and (2) No filtering, where all target items are retained, including unseen items, introducing a larger item shift. Experiments for "No filtering" are conducted on all datasets except Electronics, which, per MetaTL's public split, corresponds only to "Filtering".

To quantify item shift, we use the Jaccard index over the top-100 popular items from each domain:

$$Jaccard_{100} = \frac{|Set_S \cap Set_T|}{|Set_S \cup Set_T|}, \tag{11}$$

where $Set_S$ and $Set_S$ are the top-100 popular items in the source and target domains, respectively. Lower values indicate greater shifts in item popularity. We report Jaccard statistics in Table 1. All datasets use $max\_len = 4$ for target sequences to ensure consistent length shift. As expected, time-based splits, especially "No filtering", exhibit significantly lower Jaccard values, reflecting increased item shift. Note that the Jaccard index captures only the popularity-based aspect of item shift, not the full distributional change.

We follow Wang et al. (2021) to randomly divide the cold-start users in the target dataset into validation and test sets, with 30% of the cold-start users being allocated to the validation set. The last interaction of the sequence is used for performance evaluation and the remaining for training. Following Wang et al. (2021), we randomly sample 100 negative items per ground truth item for evaluation. The model ranks the ground truth item among these negatives, and we assess performance using standard ranking metrics: Normalized Discounted Cumulative Gain (NDCG) and Hit Ratio (HR) at $k = 1, 5, 10$, and Mean Reciprocal Rank (MRR). We use NDCG@10 for model selection.

In addition to the Amazon datasets, we conduct supplementary experiments on the MovieLens 1M (ML-1M) dataset. As genuine cold-start users cannot be obtained via timestamp-based splitting, we construct surrogate cold-start users based on sequence length (i.e., users with short interaction histories), consistent

with the random subsampling strategy used for Amazon. These results are provided as supplementary experiments, with full details reported in the Appendix H.

## 4.2 Baseline Models

To evaluate our method, we compare it with several baselines including Matrix Factorization (MF), the transformer-based SASRec Kang & McAuley (2018), and Meta-TL Wang et al. (2021). MF and SASRec are trained on combined source and target data to implicitly leverage source knowledge, while SASRec_T, trained only on target data, serves as a lower bound using SASRec model. We also include standard transfer learning (pretrained on source, fine-tuned on target), LinRec Liu et al. (2023), a linear-complexity transformer variant, and SIGMA Liu et al. (2024), a recent Mamba-based model designed for capturing both long- and short-term preferences. For contrastive learning, we adopt CL4SRec Xie et al. (2022) as a representative baseline. For domain adaptation, we include AMID Xu et al. (2024), as it does not require shared users across domains, which aligns with our non-overlapping user setting. For direct comparison, the chosen architecture of the backbone model remains the same as SASRec throughout the experiments: the latent dimension of the embedding is d = 64, the transformer model is single-head and b = 2 blocks are stacked. Detailed implementation and hyperparameter tuning of our framework as well as other baselines can be found in Appendix D.

## 4.3 Overall Performance Comparison

**Random Split (length shift).** Table 2 shows model performance under the random split ($max\_len = 4$). Comparing SASRec_T (trained only on target) with MF (trained on all data) reveals that even a transformer struggles without knowledge transfer, highlighting the value of leveraging source domain knowledge. While SASRec outperforms MF by modeling sequential patterns, MetaTL achieves similar results to SASRec in 3 out of 5 datasets but lags behind TL and our DACSR framework, likely due to its simple MLP backbone. CL4SRec basically achieves better or close performance than its SASRec backbone under random split, indicating the augmented views of the same sequence help in this setting. The TL approach, which pretrains on source and fine-tunes on target, consistently improves over SASRec and mostly over CL4SRec, showing its effectiveness in addressing length shift. Ours achieves the best performance, particularly on Electronics, Kindle, and Toys, benefiting from its fine-grained control over knowledge transfer via $\lambda$, $\beta_T$, $\beta_{SS}$, and $p$.

**Time-Based Split (compound shift).** Table 3 shows results under the time-based split for the case of "filtering". Unlike random splits, TL offers marginal or slight negative gains over SASRec in some datasets due to significant item distribution shifts over time (see Jaccard index in Table 1), limiting the generalizability of pre-trained item embeddings. Compared to the SASRec backbone without contrastive learning, CL4SRec does not consistently achieve better performance under timestamp split. In contrast, ours consistently outperforms both SASRec and TL by better managing knowledge transfer and distribution mismatch, demonstrating its robustness under compound shifts.

CL4SRec applies contrastive augmentations uniformly, using augmented views of the same sequence as positive pairs. It does not explicitly aligns long source histories with short cold-start sequences. In contrast, our DACSR frame builds simulated target sequences from source users and aligns both user-level long/short representations and source-derived user similarity, encouraging general robustness to the cold-start length gap. AMID performs substantially worse across all five datasets under both splits. A likely reason is that its cross-domain interest grouping relies on target-side representations for similarity estimation. With only four interactions per target user, these representations are highly noisy, making stable grouping difficult. Thus, although AMID is closely related to domain adaptation, its dependence on target-side similarity limits its effectiveness in our cold-start setting.

Under the more challenging time-based split for case of "No Filtering", where entirely new items may appear in the target domain, ours yields pronounced relative gains over SASRec across every metric and dataset as shown in Table 4. These consistent uplifts show that augmenting popularity weighting with similarity-aware contrastive learning makes the model far more robust to simultaneous length and item-distribution shifts.

Table 2: Performance comparison under *Random split for length shift* with $max\_len = 4$. *Impr.* denotes the percentage improvement of our method over the strongest competing baseline. Our result is selected as the best performance among the DACSR variants.

| Dataset | Metric | SASRec_T | MF | SASRec | TL | MetaTL | SIGMA | LinRec | CL4SRec | AMID | Ours | Impr. |
|---|---|---|---|---|---|---|---|---|---|---|---|---|
| Beauty | Hit@1 | 0.1290 | 0.2107 | 0.2236 | 0.2649 | 0.202 | 0.2126 | 0.2210 | 0.2146 | 0.0390 | **0.2675** | +0.98% |
| | Hit@5 | 0.2965 | 0.3778 | 0.4023 | 0.4506 | 0.380 | 0.4299 | 0.4154 | 0.4056 | 0.1309 | **0.4958** | +10.0% |
| | NDCG@5 | 0.2163 | 0.2991 | 0.3168 | 0.3624 | 0.296 | 0.3257 | 0.3223 | 0.3150 | 0.0805 | **0.3914** | +8.00% |
| | Hit@10 | 0.3960 | 0.4405 | 0.4897 | 0.5322 | 0.464 | 0.5415 | 0.5210 | 0.5042 | 0.2288 | **0.6329** | +16.9% |
| | NDCG@10 | 0.2487 | 0.3192 | 0.3450 | 0.3885 | 0.323 | 0.3616 | 0.3564 | 0.3465 | 0.1095 | **0.4357** | +12.1% |
| | MRR | 0.2189 | 0.2970 | 0.3179 | 0.3604 | 0.297 | 0.3224 | 0.3225 | 0.3142 | 0.1000 | **0.3930** | +9.05% |
| Toys | Hit@1 | 0.1015 | 0.1917 | 0.2135 | 0.2120 | 0.201 | 0.1334 | 0.2035 | 0.2152 | 0.0145 | **0.2663** | +23.7% |
| | Hit@5 | 0.2908 | 0.3543 | 0.3746 | 0.4122 | 0.396 | 0.3617 | 0.3817 | 0.3846 | 0.0896 | **0.5302** | +28.6% |
| | NDCG@5 | 0.1976 | 0.2758 | 0.2967 | 0.3172 | 0.300 | 0.2488 | 0.2973 | 0.3017 | 0.0499 | **0.4027** | +27.0% |
| | Hit@10 | 0.4225 | 0.4202 | 0.4762 | 0.5162 | 0.503 | 0.4892 | 0.4660 | 0.4741 | 0.1524 | **0.6798** | +31.7% |
| | NDCG@10 | 0.2398 | 0.2970 | 0.3293 | 0.3509 | 0.335 | 0.2901 | 0.3246 | 0.3304 | 0.0705 | **0.4510** | +28.5% |
| | MRR | 0.2033 | 0.2716 | 0.3016 | 0.3155 | 0.301 | 0.2472 | 0.2989 | 0.3033 | 0.0680 | **0.3952** | +25.3% |
| Sports | Hit@1 | 0.1276 | 0.1657 | 0.2308 | 0.2309 | 0.194 | 0.1862 | 0.2183 | 0.2323 | 0.0529 | **0.2808** | +20.9% |
| | Hit@5 | 0.2964 | 0.3519 | 0.4683 | 0.4933 | 0.415 | 0.4408 | 0.4571 | 0.4810 | 0.1406 | **0.5770** | +17.0% |
| | NDCG@5 | 0.2124 | 0.2617 | 0.3547 | 0.3690 | 0.310 | 0.3193 | 0.3452 | 0.3627 | 0.0962 | **0.4364** | +18.3% |
| | Hit@10 | 0.3947 | 0.4383 | 0.5812 | 0.6134 | 0.516 | 0.5796 | 0.5701 | 0.5961 | 0.2283 | **0.7100** | +15.7% |
| | NDCG@10 | 0.2440 | 0.2898 | 0.3912 | 0.4080 | 0.342 | 0.3642 | 0.3816 | 0.4000 | 0.1239 | **0.4795** | +17.5% |
| | MRR | 0.2141 | 0.2585 | 0.3478 | 0.3583 | 0.305 | 0.3131 | 0.3385 | 0.3532 | 0.1148 | **0.4216** | +17.7% |
| Kindle | Hit@1 | 0.0510 | 0.3054 | 0.3757 | 0.3731 | 0.357 | 0.2476 | 0.3802 | 0.4022 | 0.0217 | **0.4093** | +1.77% |
| | Hit@5 | 0.1414 | 0.5220 | 0.6259 | 0.6501 | 0.632 | 0.4217 | 0.6123 | 0.6194 | 0.0797 | **0.6581** | +1.23% |
| | NDCG@5 | 0.0950 | 0.4194 | 0.5094 | 0.5217 | 0.504 | 0.3393 | 0.4949 | 0.5176 | 0.0468 | **0.5413** | +3.76% |
| | Hit@10 | 0.2203 | 0.5918 | 0.7300 | 0.7383 | 0.742 | 0.5273 | 0.7070 | 0.7235 | 0.1386 | **0.7631** | +2.84% |
| | NDCG@10 | 0.1201 | 0.4424 | 0.5430 | 0.5503 | 0.540 | 0.3730 | 0.5253 | 0.5510 | 0.0662 | **0.5758** | +4.50% |
| | MRR | 0.1122 | 0.4045 | 0.4951 | 0.5032 | 0.487 | 0.3418 | 0.4802 | 0.5096 | 0.0672 | **0.5267** | +3.36% |
| Electronic | Hit@1 | 0.1458 | 0.1363 | 0.2163 | 0.2193 | 0.193 | 0.1913 | 0.2231 | 0.2035 | 0.0773 | **0.2448** | +9.73% |
| | Hit@5 | 0.3326 | 0.3212 | 0.4627 | 0.4871 | 0.420 | 0.4240 | 0.4827 | 0.4797 | 0.1937 | **0.5268** | +8.15% |
| | NDCG@5 | 0.2413 | 0.2337 | 0.3462 | 0.3622 | 0.310 | 0.3113 | 0.3586 | 0.3468 | 0.1335 | **0.3931** | +8.53% |
| | Hit@10 | 0.4616 | 0.4040 | 0.5878 | 0.6233 | 0.553 | 0.5666 | 0.6009 | 0.5997 | 0.2827 | **0.6818** | +9.39% |
| | NDCG@10 | 0.2828 | 0.2603 | 0.3871 | 0.4064 | 0.353 | 0.3574 | 0.3965 | 0.3861 | 0.1611 | **0.4433** | +9.08% |
| | MRR | 0.2435 | 0.2304 | 0.3406 | 0.3542 | 0.307 | 0.3109 | 0.3493 | 0.3349 | 0.1459 | **0.3846** | +8.58% |

We also evaluate full-item ranking under the time-based split, which reflects real cold-start new users. The results are consistent with our main findings and are reported in Appendix E as complementary results.

## 4.4 Ablation Study

**Comparisons for Model Variants.** Table 5 shows the performance by DACSR, DACSR+ and DACSR++. All three variants consistently outperform the SASRec baseline. By weighting the loss by item popularity in DACSR+, this yields steady, if modest, uplifts over DACSR in most ranking metrics. Building on that, DACSR++ adds a similarity-weighted contrastive component that aligns full-sequence and partial-sequence embeddings more tightly; this additional signal produces the largest gains of all, especially under the random split which presents a smaller item shift relative to time split. DACSR++ is generally the strongest overall, with Kindle being the main exception. This may be due to Kindle's weaker source-domain similarity structure, which makes similarity-guided transfer less effective than the simpler DACSR+ variant; details are provided in Appendix G.

Table 3: Model performances under *Time-based split* with compound knowledge shift. Impr. denotes the percentage improvement of our best DACSR variants over the next best method.

| Dataset | Metric | SASRec | MetaTL | TL | SIGMA | LinRec | CL4SRec | AMID | Ours | Impr. |
|---------|--------|--------|--------|------|-------|--------|---------|------|------|-------|
| Beauty | Hit@1 | 0.4190 | 0.347 | 0.3851 | 0.3882 | 0.3915 | 0.4151 | 0.1644 | **0.4229** | +0.93% |
| | Hit@5 | 0.6444 | 0.534 | 0.6478 | 0.5766 | 0.6159 | 0.6439 | 0.3324 | **0.6783** | +4.71% |
| | Hit@10 | 0.7342 | 0.638 | 0.7568 | 0.6722 | 0.7086 | 0.7377 | 0.4257 | **0.7981** | +5.46% |
| | NDCG@5 | 0.5394 | 0.443 | 0.5228 | 0.4873 | 0.5114 | 0.5365 | 0.2504 | **0.5581** | +3.47% |
| | NDCG@10 | 0.5687 | 0.477 | 0.5582 | 0.5182 | 0.5416 | 0.5668 | 0.2765 | **0.5967** | +4.92% |
| | MRR | 0.5276 | 0.441 | 0.5068 | 0.4828 | 0.5009 | 0.5235 | 0.2545 | **0.5446** | +3.22% |
| Toy | Hit@1 | 0.3345 | 0.268 | 0.3245 | 0.2848 | 0.3486 | 0.3104 | 0.0720 | **0.3807** | +9.21% |
| | Hit@5 | 0.5731 | 0.483 | 0.5863 | 0.4840 | 0.5700 | 0.5345 | 0.2246 | **0.6696** | +14.2% |
| | Hit@10 | 0.6947 | 0.601 | 0.7013 | 0.6047 | 0.6710 | 0.6474 | 0.3506 | **0.8073** | +15.1% |
| | NDCG@5 | 0.4575 | 0.379 | 0.4609 | 0.3871 | 0.4632 | 0.4267 | 0.1475 | **0.5312** | +14.7% |
| | NDCG@10 | 0.4967 | 0.417 | 0.4981 | 0.4261 | 0.4955 | 0.4632 | 0.1814 | **0.5763** | +15.7% |
| | MRR | 0.4484 | 0.375 | 0.4478 | 0.3860 | 0.4542 | 0.4210 | 0.1581 | **0.5144** | +13.3% |
| Sports | Hit@1 | 0.3026 | 0.257 | 0.3114 | 0.2573 | 0.3101 | 0.3131 | 0.1034 | **0.3572** | +14.1% |
| | Hit@5 | 0.5753 | 0.517 | 0.6118 | 0.5139 | 0.5683 | 0.6039 | 0.2540 | **0.6864** | +12.2% |
| | Hit@10 | 0.6916 | 0.634 | 0.7276 | 0.6328 | 0.6683 | 0.7014 | 0.3620 | **0.8106** | +11.4% |
| | NDCG@5 | 0.4483 | 0.393 | 0.4701 | 0.3904 | 0.4445 | 0.4669 | 0.1771 | **0.5307** | +12.9% |
| | NDCG@10 | 0.4859 | 0.431 | 0.5075 | 0.4287 | 0.4768 | 0.4985 | 0.2115 | **0.5710** | +12.5% |
| | MRR | 0.4346 | 0.382 | 0.4497 | 0.3800 | 0.4302 | 0.4472 | 0.1877 | **0.5053** | +12.4% |
| Kindle | Hit@1 | 0.7223 | 0.678 | 0.8171 | 0.6795 | 0.7232 | 0.8060 | 0.1206 | **0.8489** | +3.89% |
| | Hit@5 | 0.9719 | 0.932 | 0.9687 | 0.9286 | 0.9565 | 0.9701 | 0.2691 | **0.9857** | +1.42% |
| | Hit@10 | 0.9891 | 0.969 | 0.9875 | 0.9730 | 0.9875 | 0.9922 | 0.3527 | **0.9974** | +0.52% |
| | NDCG@5 | 0.8606 | 0.817 | 0.9064 | 0.8179 | 0.8477 | 0.8979 | 0.1896 | **0.9281** | +2.39% |
| | NDCG@10 | 0.8660 | 0.829 | 0.9127 | 0.8322 | 0.8580 | 0.9052 | 0.2207 | **0.9320** | +2.11% |
| | MRR | 0.8256 | 0.785 | 0.8883 | 0.7879 | 0.8163 | 0.8769 | 0.2042 | **0.9101** | +2.45% |
| Electr | Hit@1 | 0.2727 | 0.225 | 0.2697 | 0.2048 | 0.2558 | 0.2538 | 0.1828 | **0.2966** | +8.76% |
| | Hit@5 | 0.5505 | 0.482 | 0.5509 | 0.4475 | 0.5154 | 0.5181 | 0.3836 | **0.5837** | +5.95% |
| | Hit@10 | 0.6679 | 0.602 | 0.6680 | 0.5846 | 0.6402 | 0.6466 | 0.4987 | **0.6991** | +4.66% |
| | NDCG@5 | 0.4186 | 0.359 | 0.4179 | 0.3297 | 0.3925 | 0.3922 | 0.2866 | **0.4479** | +7.00% |
| | NDCG@10 | 0.4568 | 0.398 | 0.4557 | 0.3740 | 0.4331 | 0.4337 | 0.3182 | **0.4854** | +6.26% |
| | MRR | 0.4050 | 0.351 | 0.4041 | 0.3260 | 0.3838 | 0.3829 | 0.2859 | **0.4323** | +6.74% |

Table 4: Performances of models under *Time-based split for both length and item shifts* when $max\_len = 4$ for case of "No Filtering". Electronic data is from Wang et al. (2021) and all items in the target are observed in the source. Therefore, no result of Electronic are reported in this case.

| Metrics | Model | Beauty | Impr. | Toy | Impr. | Sports | Impr. | Kindle | Impr. |
|---------|-------|--------|-------|-----|-------|--------|-------|--------|-------|
| Hit@1 | SASRec | 0.3443 | +17.3% | 0.3172 | +11.2% | 0.2784 | +12.0% | 0.6549 | +10.2% |
| | DACSR++ | **0.4039** | | **0.3526** | | **0.3118** | | **0.7218** | |
| Hit@5 | SASRec | 0.5409 | +11.0% | 0.5172 | +11.3% | 0.5124 | +9.1% | 0.8927 | +6.5% |
| | DACSR++ | **0.6005** | | **0.5754** | | **0.5590** | | **0.9508** | |
| Hit@10 | SASRec | 0.6200 | +10.7% | 0.6222 | +9.0% | 0.6120 | +8.9% | 0.9290 | +5.4% |
| | DACSR++ | **0.6863** | | **0.6783** | | **0.6663** | | **0.9793** | |
| NDCG@5 | SASRec | 0.4485 | +13.0% | 0.4216 | +11.1% | 0.3987 | +10.9% | 0.7855 | +8.4% |
| | DACSR++ | **0.5067** | | **0.4685** | | **0.4424** | | **0.8518** | |
| NDCG@10 | SASRec | 0.4742 | +12.7% | 0.4555 | +10.1% | 0.4312 | +10.6% | 0.7973 | +8.0% |
| | DACSR++ | **0.5343** | | **0.5016** | | **0.4769** | | **0.8610** | |
| MRR | SASRec | 0.4410 | +13.0% | 0.4163 | +10.1% | 0.3874 | +10.8% | 0.7570 | +8.7% |
| | DACSR++ | **0.4986** | | **0.4583** | | **0.4294** | | **0.8228** | |

**Performance against Different $max\_len$.** The parameter $max\_len$ controls the maximum sequence length in the target data to simulate cold-start scenarios. To evaluate the impact of length shift, we vary $max\_len \in \{4, 6, 10, 20\}$ under the *Random split* and compare DACSR++ with SASRec in Table 6,

Table 5: Performance of DACSR variants under two splits, with $max\_len = 4$.

| Dataset | Metric | Random split | | | Time split | | |
|---|---|---|---|---|---|---|---|
| | | DACSR | DACSR+ | DACSR++ | DACSR | DACSR+ | DACSR++ |
| Beauty | Hit@1 | 0.2548 | 0.2563 | **0.2675** | **0.4355** | 0.4329 | 0.4229 |
| | Hit@5 | 0.4342 | 0.4374 | **0.4958** | 0.6605 | 0.6552 | **0.6783** |
| | NDCG@5 | 0.3486 | 0.3507 | **0.3914** | 0.5560 | 0.5530 | **0.5581** |
| | Hit@10 | 0.5317 | 0.5375 | **0.6329** | 0.7586 | 0.7599 | **0.7981** |
| | NDCG@10 | 0.3802 | 0.3830 | **0.4357** | 0.5878 | 0.5866 | **0.5967** |
| | MRR | 0.3500 | 0.3529 | **0.3930** | 0.5444 | 0.5434 | **0.5446** |
| Toys | Hit@1 | 0.2518 | 0.2611 | **0.2663** | 0.3748 | 0.3758 | **0.3807** |
| | Hit@5 | 0.4270 | 0.4454 | **0.5302** | 0.6046 | 0.6261 | **0.6696** |
| | NDCG@5 | 0.3440 | 0.3586 | **0.4027** | 0.4949 | 0.5100 | **0.5312** |
| | Hit@10 | 0.5167 | 0.5354 | **0.6798** | 0.7133 | 0.7343 | **0.8073** |
| | NDCG@10 | 0.3727 | 0.3877 | **0.4510** | 0.5298 | 0.5448 | **0.5763** |
| | MRR | 0.3449 | 0.3582 | **0.3952** | 0.4853 | 0.4977 | **0.5144** |
| Sports | Hit@1 | 0.2516 | 0.2477 | **0.2808** | 0.3396 | 0.3512 | **0.3572** |
| | Hit@5 | 0.4873 | 0.4727 | **0.5770** | 0.6100 | 0.6115 | **0.6864** |
| | NDCG@5 | 0.3750 | 0.3666 | **0.4364** | 0.4815 | 0.4875 | **0.5307** |
| | Hit@10 | 0.5890 | 0.5899 | **0.7100** | 0.7145 | 0.7289 | **0.8106** |
| | NDCG@10 | 0.4082 | 0.4046 | **0.4795** | 0.5153 | 0.5258 | **0.5710** |
| | MRR | 0.3668 | 0.3626 | **0.4216** | 0.4646 | 0.4742 | **0.5053** |
| Kindle | Hit@1 | 0.3981 | **0.4093** | 0.3647 | 0.8203 | **0.8489** | 0.8281 |
| | Hit@5 | 0.6535 | 0.6581 | **0.6621** | 0.9797 | **0.9857** | 0.9828 |
| | NDCG@5 | 0.5302 | **0.5413** | 0.5250 | 0.9114 | **0.9281** | 0.9160 |
| | Hit@10 | 0.7535 | **0.7631** | 0.7519 | 0.9938 | **0.9974** | 0.9922 |
| | NDCG@10 | 0.5627 | **0.5758** | 0.5540 | 0.9160 | **0.9320** | 0.9191 |
| | MRR | 0.5130 | **0.5267** | 0.5016 | 0.8905 | **0.9101** | 0.8951 |
| Electr | Hit@1 | 0.2433 | 0.2439 | **0.2448** | 0.2900 | **0.2983** | 0.2966 |
| | Hit@5 | 0.4969 | 0.5055 | **0.5268** | 0.5695 | 0.5716 | **0.5837** |
| | NDCG@5 | 0.3755 | 0.3794 | **0.3931** | 0.4375 | 0.4421 | **0.4479** |
| | Hit@10 | 0.6175 | 0.6183 | **0.6818** | 0.6850 | 0.6903 | **0.6991** |
| | NDCG@10 | 0.4144 | 0.4159 | **0.4433** | 0.4748 | 0.4803 | **0.4854** |
| | MRR | 0.3654 | 0.3672 | **0.3846** | 0.4228 | 0.4284 | **0.4323** |

reporting DACSR++'s relative improvement. Results show that most models benefit from longer target sequences, confirming that longer interaction improves prediction. DACSR++ generally outperforms others, especially when $max\_len$ is small, highlighting DACSR++'s advantage under severe length shift. The gap generally narrows as target sequences grow longer, but not uniformly across every model/dataset pair.

**Impact of the Contrastive Losses.** The user representations using T-SNE have been plotted in Figure 2, where SASRec yields two nearly disjoint clusters of source and target embeddings under both time-based and random splits, reflecting a pronounced domain gap and limited representation alignment. In contrast, DACSR+ achieves greater overlap between source and target, and DACSR++ further enhances this alignment, producing the most coherent intermixing of domains. DACSR++ exhibits the tightest intra-target clustering, which can be attributed to its similarity-weighted contrastive loss: by pulling together semantically related user representations rather than treating all non-paired examples as strict negatives, it fosters more meaningful grouping of target users than the standard DACSR+ formulation. The TSNE plots for other datasets are provided in Appendix F, showing similar results.

Ablation studies comparing DACSR variants with and without CL are provided, where we evaluate the impact of contrastive loss under both split settings with $max\_len = 4$. Setting $\lambda = 0$ removes the contrastive

Table 6: Performances of models for *Random split* against different *max_len*. NDCG@10 values are reported. Impr. is computed against the strongest competing baseline for each dataset and target length.

| *max_len* | Model | Electr | Impr. | Kindle | Impr. | Beauty | Impr. | Toy | Impr. | Sports | Impr. |
|---|---|---|---|---|---|---|---|---|---|---|---|
| 20 | DACSR++ | 0.4824 | | 0.6272 | | **0.4368** | | **0.4214** | | **0.4373** | |
| | SASRec | 0.4817 | | **0.6276** | | 0.3763 | | 0.3778 | | 0.3581 | |
| | TL | **0.4840** | -0.33% | 0.6151 | -0.06% | 0.3748 | +16.08% | 0.3567 | +11.54% | 0.3524 | +22.12% |
| | SIGMA | 0.4246 | | 0.4104 | | 0.3680 | | 0.3005 | | 0.2911 | |
| | LinRec | 0.4685 | | 0.6060 | | 0.3744 | | 0.3549 | | 0.3524 | |
| 10 | DACSR++ | **0.4811** | | **0.6360** | | **0.4566** | | **0.4434** | | **0.4717** | |
| | SASRec | 0.4510 | | 0.6275 | | 0.3720 | | 0.3468 | | 0.3744 | |
| | TL | 0.4526 | +6.30% | 0.6132 | +1.35% | 0.3675 | +21.47% | 0.3489 | +27.08% | 0.3749 | +25.82% |
| | SIGMA | 0.4031 | | 0.3450 | | 0.3738 | | 0.2854 | | 0.3269 | |
| | LinRec | 0.4433 | | 0.5811 | | 0.3759 | | 0.3482 | | 0.3595 | |
| 6 | DACSR++ | **0.4879** | | **0.6173** | | **0.4517** | | **0.4314** | | **0.4847** | |
| | SASRec | 0.4241 | | 0.5808 | | 0.3714 | | 0.3236 | | 0.3833 | |
| | TL | 0.4016 | +10.71% | 0.6007 | +2.76% | 0.3747 | +19.76% | 0.2875 | +28.97% | 0.4128 | +17.42% |
| | SIGMA | 0.4117 | | 0.3651 | | 0.3751 | | 0.2887 | | 0.3622 | |
| | LinRec | 0.4407 | | 0.5812 | | 0.3772 | | 0.3345 | | 0.3739 | |
| 4 | DACSR++ | **0.4433** | | **0.5540** | | **0.4357** | | **0.4510** | | **0.4795** | |
| | SASRec | 0.3871 | | 0.5430 | | 0.3450 | | 0.3294 | | 0.3911 | |
| | TL | 0.4064 | +9.08% | 0.5503 | +0.67% | 0.3885 | +12.15% | 0.3509 | +28.53% | 0.4080 | +17.52% |
| | SIGMA | 0.3574 | | 0.3730 | | 0.3616 | | 0.2901 | | 0.3642 | |
| | LinRec | 0.3965 | | 0.5253 | | 0.3564 | | 0.3246 | | 0.3816 | |

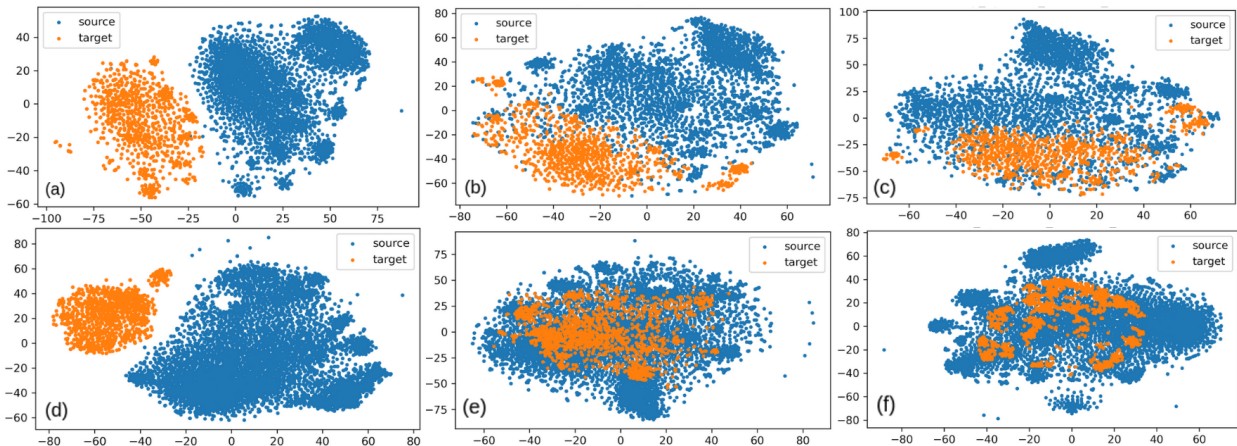

Figure 2: TSNE visualization of *Toys*'s *source* (blue) and *target* (orange) user representations under two splitting strategies: *time-split* (a, b, c) and *random-split* (d, e, f) for three models: *SASRec* (a, d), *DACSR+* (b, e), and *DACSR++* (c, f). Improved mixing of target and source clusters indicates more effective domain alignment.

loss and the emulated target domain, reducing DACSR to two independently trained transformers with shared item embeddings and only balanced by $\beta_T$. As shown in Table 7, the contrastive loss significantly improves performance across both split methods. The gains are more pronounced under the *Random split*, where length shift dominates, allowing more effective knowledge transfer via the emulated domain. Under the *Time-based split*, the added complexity from item distribution shift limits the extent of information flow, but improvements are still evident.

Table 7: Performances with or without contrastive loss when $max\_len = 4$. NDCG@10 values are reported.

| Splits | Length shift | Item shift | Model | Electr | Kindle | Beauty | Toy | Sports |
|--------|:---:|:---:|--------|--------|--------|--------|--------|--------|
| *Time-based* | ✓ | ✓ | DACSR w/o $L_\delta$ | 0.4469 | 0.9104 | 0.5721 | 0.4966 | 0.5006 |
| | | | DACSR | 0.4748 | 0.9160 | 0.5878 | 0.5298 | 0.5153 |
| | | | DACSR+ | 0.4803 | **0.9320** | 0.5866 | 0.5448 | 0.5258 |
| | | | DACSR++ | **0.4854** | 0.9191 | **0.5967** | **0.5763** | **0.5710** |
| *Random* | ✓ | ✗ | DACSR w/o $L_\delta$ | 0.3522 | 0.4469 | 0.3359 | 0.2629 | 0.3675 |
| | | | DACSR | 0.4144 | 0.5627 | 0.3804 | 0.3727 | 0.4082 |
| | | | DACSR+ | 0.4159 | **0.5758** | 0.3830 | 0.3877 | 0.4046 |
| | | | DACSR++ | **0.4433** | 0.5540 | **0.4357** | **0.4510** | **0.4795** |

# 5   Computational Complexity

We analyze the computational and parameter overhead of DACSR++ compared to a standard SASRec model.

**In-batch Similarity.** Within each training batch of size $B$, DACSR++ computes a pairwise similarity matrix over $d$-dimensional user representations. This requires $B^2$ dot products, each of dimension $d$, resulting in a per-batch cost of $O(B^2 d)$. Over an epoch with $N$ users (i.e., $N/B$ batches), the total cost becomes $O(NBd)$, which scales linearly with the number of users $N$ since $B$ and $d$ are fixed.

**Model Parameters and Training Computation.** DACSR++ adopts a dual-encoder architecture with shared item embeddings. Let $V$ denote the vocabulary size, $d$ the hidden dimension, $b$ the number of transformer blocks, $L_s$ and $L_t$ the source and target sequence lengths, and $B$ the batch size. Following standard Transformer analysis Vaswani et al. (2017), the cost per block is $O(Ld^2)$ for projections and feed-forward layers and $O(L^2 d)$ for self-attention. Table 8 summarizes both model size and per-step training complexity.

Table 8: Model size and computational complexity of DACSR++ versus SASRec.

| | Component | SASRec | DACSR++ |
|--------|-----------|--------|---------|
| *Parameters* | Item embedding | $Vd$ | $Vd$ (shared) |
| | Transformer layers | $O(bd^2)$ | $O(2bd^2)$ |
| *Training (per step)* | Encoder | $O(Bb(Ld^2 + L^2 d))$ | $O(Bb(L_s d^2 + L_s^2 d))$ |
| | Target encoder (pseudo + real) | — | $O(2Bb(L_t d^2 + L_t^2 d))$ |
| | Similarity matrix | — | $O(B^2 d)$ |

Since the embedding table dominates when $V \gg bd$, the additional parameter overhead amounts to one extra transformer stack and remains modest. The dominant training cost is the target encoder pass, which runs twice to handle both pseudo and real target sequences.

**Empirical Overhead.** On a representative dataset, DACSR++ introduces a modest overhead compared to SASRec: parameter count increases by approximately 8%, and training time per epoch increases by about 16%. This overhead stems from the additional encoder pass and similarity computation, both standard in dual-encoder and contrastive frameworks. At inference time, only the target encoder is used for recommendation. The source encoder and similarity computation are not involved, resulting in identical inference cost to SASRec.

## 6 Limitations

In this section, we clarify the roles of the two evaluation protocols. While we report results under both random and time-based splits, the two serve different purposes. The time-based split reflects genuine cold-start behavior, where target users are new users with short interaction sequences, and is therefore the primary evaluation setting. In contrast, the random split provides a controlled comparison but does not simulate true cold-start conditions. Furthermore, we clarify that the cold-start scenario under the time-based split is validated only on Amazon datasets. MovieLens-1M does not yield genuine new users under a temporal split due to its static snapshot nature, and thus its results correspond to a controlled short-history setting rather than a strict temporal cold-start scenario.

A second limitation concerns hyperparameter overhead. Although the per-step training and inference costs of DACSR++ are comparable to SASRec, the model introduces additional hyperparameters such as loss weights and masking probability. Consequently, the total wall-clock cost including hyperparameter tuning can be several times higher than a single SASRec run.

## 7 Conclusion

We tackle cold-start sequential recommendation via domain adaptation by addressing two key shifts: length shift and compound length-item distribution shift. Our main contribution is a dual-transformer framework with separate models for long (source) and short (target) sequences, using balanced joint training to mitigate bias and enable adaptive knowledge transfer. We emulate a cold-start target domain by sampling short sequences from source data, enriching the target data diversity under length shift. For the compound shift, we reduce item bias with popularity-weighted user representations and enhance transfer via contrastive learning based on user similarity. Extensive experiments demonstrate that our approach consistently outperforms state-of-the-art methods across both shift scenarios.

## 8 Acknowledgment

We are grateful to Prof. Li Xiaoli and Dr. Ramon Sagarna for their insightful discussions and valuable feedback, which greatly benefited this work.

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

## A    Training Strategy

For DACSR++, we introduce an epoch-dependent weight $\alpha(e)$ defined as

$$\alpha(e) = \begin{cases} 1.0, & e < E_{\text{warmup}} = 15, \\ 1 - \dfrac{\min\big(e,\, E_{\text{decay}}\big)}{E_{\text{decay}}}, & e \geq E_{\text{warmup}}, \end{cases}$$

where $e$ is the epoch index, $E_{\text{decay}}$ is a tunable hyperparameter and $E_{\text{warmup}}$ denotes the number of initial training epochs, during which the model uses the standard in-batch contrastive loss. This yields the overall objective:

$$L_\delta(e) \;=\; \alpha(e)\, L_\delta^{in} \;+\; \big(1 - \alpha(e)\big)\, L_\delta^{s}.$$

By adjusting $E_{\text{decay}}$, we control the rate at which the model transitions from the robust in-batch loss to the finer-grained similarity-weighted variant as the learned similarity estimates become reliable.

## B    Hyperparameters and Their Impacts

The overall loss function to be minimized is then

$$L = L_{pred,S} + \beta_{SS} L_{pred,SS} + \beta_T L_{pred,T} + \lambda L_\delta, \tag{12}$$

where $\beta_{SS}$, $\beta_T$ , $\lambda$ are hyperparameters, $L_{pred,S}$, $L_{pred,SS}$ and $L_{pred,T}$ are the CE loss of source, emulated target and target, respectively, and $L_\delta$ is the contrastive loss.

The elements outlined above, together with the sampling probability $p$ for the augmentation operator, collectively constitute the hyperparameters of our framework. These hyperparameters affect the characteristics of the emulated target domain, as well as balance the learning focus placed on each domain. Depending on the specific configurations, the framework is tailored to address particular aspects of the knowledge transfer process.

For example, if $p$ is close to 1, the target network will take short and long sequences, and we therefore try to align the latter, at context level, with similar sequences from the source model. By contrast, if $p$ is relatively low, the lengths of the emulated target and target sequences may be similar. The probability $p$ assumes a role in regulating the extent of information needed to be aligned between the emulated target context and the source model's context.

The alignment of contexts between the source and emulated target (and target) domains is obtained by the contrastive loss of $L_\delta$. Importantly, the hyperparameter $\lambda$ is utilized to regulate the extent to which the alignment between the two domains should be. Hyperparameter $\lambda$ together with sampling probability $p$

control the alignment process. Moderating $p$ and $\lambda$ allows us to adjust the extent of knowledge shared and transferred through source to target. The hyperparameters are tuned according to the model's performance on validation data.

## C Statistics of Datasets

The statistics of the datasets after preprocessing using the two split methods are summarized in Table 9. The *Random split for length shift* maintains substantial similarity in item distribution between source and target after preprocessing, resulting in minimal item shift. In contrast, the *Time-based split* introduces a compound shift in both sequence length and item distribution.

Table 9: Dataset statistics after preprocessing under two split strategies. *Random split for length shift* (top) produces minimal item shift between source and target. *Time-based split for compound length and item shifts* (bottom) introduces distributional shift in both sequence length and item overlap.

| Dataset | Source | | | | | | | Target | | | | | | |
|---|---|---|---|---|---|---|---|---|---|---|---|---|---|---|
| | #users | #items | Avg | min | 25% | 50% | 75% | #users | #items | Avg | min | 25% | 50% | 75% |
| *Random split for length shift* | | | | | | | | | | | | | | |
| Kindle | 6448 | 21220 | 21.5 | 10 | 12 | 15 | 23 | 1612 | 13770 | 20.7 | 6 | 11 | 15 | 23 |
| Electronics | 8651 | 41822 | 16.2 | 10 | 11 | 13 | 17 | 2153 | 16935 | 13.1 | 4 | 9 | 10 | 14 |
| Beauty | 6423 | 47432 | 17.4 | 10 | 11 | 13 | 18 | 1532 | 17607 | 12.9 | 4 | 7 | 10 | 14 |
| Sports | 10965 | 73399 | 16.2 | 10 | 11 | 13 | 27 | 2625 | 27109 | 12.2 | 4 | 8 | 10 | 13 |
| Toys | 8038 | 73114 | 18.0 | 10 | 11 | 13 | 18 | 1855 | 26470 | 12.6 | 4 | 7 | 10 | 14 |
| *Time-based split for compound length and item shifts* | | | | | | | | | | | | | | |
| Kindle | 16465 | 21220 | 25.2 | 10 | 12 | 16 | 27 | 983 | 871 | 10.2 | 4 | 5 | 7 | 12 |
| Electronics | 20865 | 20712 | 17.3 | 10 | 11 | 13 | 18 | 5963 | 9374 | 5.7 | 4 | 4 | 5 | 6 |
| Beauty | 3653 | 14841 | 17.3 | 10 | 11 | 13 | 18 | 2172 | 17607 | 6.1 | 4 | 4 | 5 | 6 |
| Sports | 6493 | 22995 | 15.6 | 10 | 11 | 13 | 17 | 2010 | 6716 | 5.1 | 4 | 4 | 4 | 6 |
| Toys | 3984 | 18052 | 17.2 | 10 | 11 | 13 | 18 | 1829 | 4665 | 4.8 | 4 | 4 | 4 | 5 |

## D Implementation Details

The chosen architecture of the backbone model remains the same throughout the experiments: the latent dimension of the embedding is $d = 64$, the transformer model is single-head and $b = 2$ blocks are stacked. For training, we set the batch size to 128, and use the Adam optimizer with a learning rate of 0.001. Network weights of all the models are initialized with a normal-distribution with a standard deviation of 0.01. We use the validation performance in the target domain for early stopping when there is no performance improvement over 20 training epochs. Based on the dataset statistics in Table 9, we set the sequence length $L$ of both backbone models to be 30 for Kindle dataset and 20 for the other four datasets. The values considered for other hyperparameters of the backbone model and the DACSR++ framework are summarized in Table 10. These hyperparameters are set through the following empirical grid search approach:

- First, we train a transformer model SASRec on the combined source and target data. Note that this model also serves as a baseline in our evaluation. The optimal model and embedding dropout rates are searched over their respective ranges. The resulting model, along with the determined dropout rate, is used as the backbone for both source and target network in DACSR++.
- With the backbone models established, we set $p$ to a fixed value of 0.5. Using this fixed value, we search for the optimal values of the domain information balance control parameter, i.e., $\beta_T$, $\beta_{SS}$ and the contrastive loss control parameter, i.e., $\lambda$ within their respective ranges.
- Finally, with optimal $\beta_T$, $\beta_{SS}$ and $\lambda$ values determined, we subsequently tune the sampling probability $p$.

To ensure fair comparison, CL4SRec was tuned in two stages using the official code: (1) first optimizing the SASRec backbone (trained on combined source and target data without contrastive loss) for model

Table 10: Hyperparameters and their sets of values.

| hyperparameter | Searching Values |
|---|---|
| Model dropout rate | 0.1, 0.2, 0.3, 0.4, 0.5 |
| Embedding dropout rate | 0.1, 0.2, 0.3, 0.4, 0.5 |
| $\beta_T$ | 0, 0.01, 0.03, 0.04, 0.05, 0.07, 0.1, 0.2, 0.3 |
| $\beta_{SS}$ | 0, 0.01, 0.03, 0.05, 0.07, 0.1 |
| $\lambda$ | 0, 0.01, 0.03, 0.04, 0.05, 0.07, 0.1, 0.2, 0.3 |
| $p$ | 0.3, 0.4, 0.5, 0.6, 0.7, 0.8 |

and embedding dropout using the same grid as the SASRec baseline, preventing the contrastive loss from dominating early training; and (2) then searching CL-specific hyperparameters including contrastive loss weight (0.05, 0.1, 0.2), temperature (0.2, 0.5, 1.0, 2.0), and similarity, on the validation set with backbone dropout fixed from Stage 1. For AMID, we use the official repository with its full Inner/Inter similarity and debiasing branches. To adapt the model to our source-target setting, where source and target users are disjoint rather than overlapping, we tune the enabling of the Inner/Inter similarity components (four on/off combinations) on the validation set, with all other parameters set to their defaults. We do not perform extensive hyperparameter tuning for AMID given its consistently poor performance across all five datasets under our cold-start setting, where target users have only 4 interactions, too few for AMID's cross-domain interest grouping to form stable representations.

# E   Performance Evaluation with Full-item ranking

To avoid any inconsistency of the 100-negative sample evaulation to the full-ranking results, we will present additional full-ranking results to strengthen the main claims. We re-evaluate the full-ranking of the saved best checkpoints on all five datasets for four models for time-split experiments: DACSR++, SIGMA, LinRec and TL shown in Table 11, where the best result is in bold, the second-best is underlined, and Impr. is the percentage difference between DACSR++ and the strongest competing baseline in that row. A negative Impr. means DACSR++ is below the best baseline.

These full-ranking results support two key observations. First, the improvements are not merely artifacts of the 100-negative evaluation protocol. Second, performance under full ranking is more nuanced, rather than uniformly superior across all metrics. To further validate this beyond re-evaluating existing checkpoints, we conducted additional experiments with full-ranking training on the Toy dataset. After tuning the models, the best-performing configurations of DACSR++ and LinRec under full evaluation are reported in Table 12, where DACSR++ can still outperform LinRec by a lift of 6% in NDCG@10.

# F   User Representations of Model Variants

The T-SNE visualization of user representations on the Sports dataset is shown in Figure 3, supporting conclusions similar to those from Figure 2. SASRec produces two nearly disjoint clusters of source and target embeddings under both time-based and random splits, indicating a pronounced domain gap and limited representation alignment. In contrast, DACSR+ increases the overlap between source and target, while DACSR++ further improves alignment, achieving the most coherent intermixing of domains. Notably, DACSR++ also demonstrates the tightest intra-target clustering, which can be attributed to its similarity-weighted contrastive loss: by pulling together semantically related user representations instead of treating all non-paired examples as strict negatives, it fosters more meaningful groupings of target users compared to the standard DACSR+ formulation.

# G   User Similarity Analysis over Different Datasets

We further analyze user similarity patterns across datasets, as DACSR++ explicitly exploits similarity signals derived from the source domain. We compute pairwise cosine similarity between user embeddings in the

Table 11: Time-based split full-ranking results over the full items.

| Dataset | Metric | DACSR++ | SIGMA | LinRec | TL | Impr. |
|---------|--------|---------|-------|--------|-----|-------|
| Beauty | Hit@1 | **0.0465** | 0.0395 | 0.0371 | 0.0200 | +17.72% |
| | Hit@5 | **0.1489** | 0.1329 | 0.1331 | 0.1146 | +11.87% |
| | Hit@10 | **0.2106** | 0.2028 | 0.1883 | 0.1863 | +3.85% |
| | NDCG@5 | **0.1005** | 0.0874 | 0.0867 | 0.0682 | +14.99% |
| | NDCG@10 | **0.1201** | 0.1099 | 0.1044 | 0.0917 | +9.28% |
| | MRR | **0.1032** | 0.0901 | 0.0880 | 0.0723 | +14.54% |
| Toy | Hit@1 | 0.0613 | 0.0599 | **0.0768** | 0.0632 | -20.18% |
| | Hit@5 | 0.1294 | 0.1238 | **0.1415** | 0.1138 | -8.55% |
| | Hit@10 | 0.1634 | 0.1465 | **0.1727** | 0.1509 | -5.39% |
| | NDCG@5 | 0.0962 | 0.0948 | **0.1102** | 0.0913 | -12.70% |
| | NDCG@10 | 0.1073 | 0.1022 | **0.1205** | 0.1034 | -10.95% |
| | MRR | 0.0976 | 0.0930 | **0.1107** | 0.0952 | -11.83% |
| Sports | Hit@1 | **0.0241** | 0.0195 | 0.0203 | 0.0189 | +18.72% |
| | Hit@5 | **0.0684** | 0.0535 | 0.0599 | 0.0586 | +14.19% |
| | Hit@10 | **0.1017** | 0.0730 | 0.0973 | 0.0876 | +4.52% |
| | NDCG@5 | **0.0478** | 0.0366 | 0.0405 | 0.0395 | +18.02% |
| | NDCG@10 | **0.0587** | 0.0427 | 0.0524 | 0.0486 | +12.02% |
| | MRR | **0.0535** | 0.0387 | 0.0454 | 0.0438 | +17.84% |
| Kindle | Hit@1 | **0.1635** | 0.0965 | 0.1445 | 0.1464 | +11.68% |
| | Hit@5 | **0.3583** | 0.2181 | 0.2739 | 0.2922 | +22.62% |
| | Hit@10 | **0.4240** | 0.2819 | 0.3515 | 0.3818 | +11.05% |
| | NDCG@5 | **0.2630** | 0.1603 | 0.2115 | 0.2179 | +20.70% |
| | NDCG@10 | **0.2843** | 0.1807 | 0.2362 | 0.2458 | +15.66% |
| | MRR | **0.2546** | 0.1641 | 0.2132 | 0.2189 | +16.31% |
| Electr. | Hit@1 | 0.0126 | 0.0083 | 0.0124 | **0.0136** | -7.35% |
| | Hit@5 | **0.0562** | 0.0283 | 0.0453 | 0.0473 | +18.82% |
| | Hit@10 | **0.0820** | 0.0471 | 0.0683 | 0.0751 | +9.19% |
| | NDCG@5 | **0.0340** | 0.0184 | 0.0287 | 0.0304 | +11.84% |
| | NDCG@10 | **0.0424** | 0.0244 | 0.0362 | 0.0393 | +7.89% |
| | MRR | **0.0369** | 0.0226 | 0.0323 | 0.0348 | +6.03% |

Table 12: Finetuned Time-based split full-ranking result of dataset toy.

| Model | Recall@1 | Recall@5 | Recall@10 | NDCG@1 | NDCG@5 | NDCG@10 | MRR |
|-------|----------|----------|-----------|--------|--------|---------|-----|
| DACSR++ | 0.0924 | 0.1695 | 0.2046 | 0.0924 | 0.1316 | 0.1430 | 0.1319 |
| LinRec | 0.0774 | 0.1609 | 0.1975 | 0.0774 | 0.1228 | 0.1348 | 0.1218 |
| Impr. | +19.31% | +5.34% | +3.56% | +19.31% | +7.13% | +6.06% | +8.24% |

source domain and examine their distributions. Figure 4 presents the distributions across datasets for direct comparison, and the corresponding summary statistics of similarity mean, median and standard derivations are reported in Table 13, where all data show similar standard derivations with kindle the smallest mean and median.

Table 13: Statistics of pairwise source-user similarity across datasets.

| | Beauty | Toy | Sport | Kindle | Electronics |
|---|--------|-----|-------|--------|-------------|
| Mean | 0.1726 | 0.1479 | 0.1519 | 0.0737 | 0.1697 |
| Std | 0.2427 | 0.2405 | 0.2672 | 0.2530 | 0.2690 |
| Median | 0.1121 | 0.0977 | 0.0999 | 0.0389 | 0.1261 |

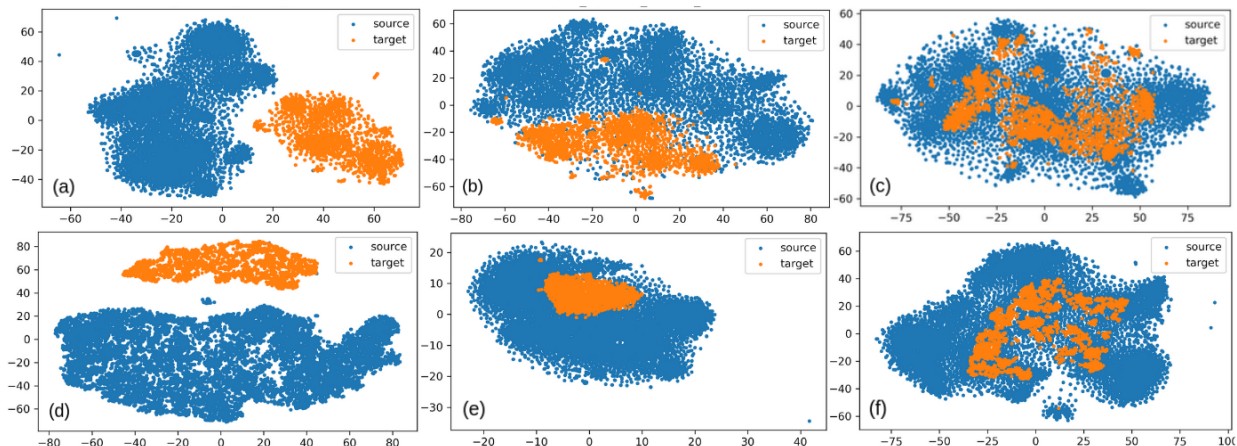

Figure 3: TSNE visualization of *Sports* dataset's *source* (blue) and *target* (orange) user embeddings under two data-splitting strategies — *by timestamp* (a, b, c) and *by random* (d, e, f) — for three models: *SASRec* (a, d), *DACSR+* (b, e), and *DACSR++* (c, f). Improved mixing of target and source clusters indicates more effective domain alignment.

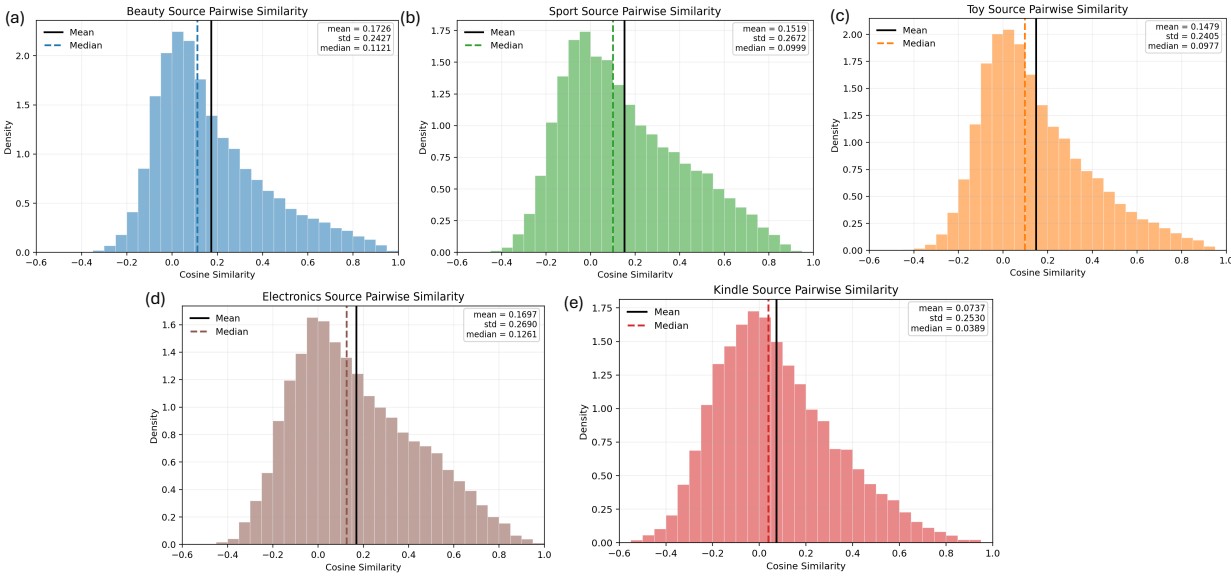

Figure 4: Source-domain user similarity distributions, computed as cosine similarities between learned user embeddings, for (a) Beauty, (b) Sports, (c) Toy, (d) Electronics, and (e) Kindle.

From Figure 4, Kindle has its similarity distribution shifted toward lower values. This suggests a weaker underlying similarity structure in the source domain, making similarity-guided transfer less effective. In the case of Kindle, the weaker similarity structure may limit the effectiveness of similarity-guided transfer, which may explain why the simpler DACSR+ variant performs comparatively better.

# H   Complementary Results on ML-1M

We further include MovieLens-1M (ML-1M) as a non-Amazon benchmark and report results for all methods under `max_len` = 4. ML-1M is a classic movie rating dataset collected by the GroupLens research lab from the MovieLens platform. Compared with the Amazon datasets, ML-1M comes from a different platform and has substantially denser user–item interaction histories, providing a complementary evaluation setting.

### H.1 Split Design and Dataset Characteristics

A strict temporal cold-start split is not suitable for ML-1M, because the dataset is a static collaborative filtering snapshot rather than a continuously evolving e-commerce log. Using ratings $\geq 4$ as positive interactions, we examine how many of the 6,038 users first appear after various timestamp cutoffs. As shown in Table 14, almost all users appear in the early stage: only 343 users (5.7%) first interact after Dec 2000, dropping to fewer than 10 beyond Jun 2001. Moreover, these 343 users are not genuinely cold-start — their average interaction length is 70.5, with 267 out of 343 having 20+ interactions. A timestamp split on ML-1M therefore separates early and late registrants rather than sparse and dense users.

Table 14: Left: new users after each timestamp cutoff on ML-1M. Right: interaction-length distribution of the 343 users first appearing after Dec 2000.

| Cutoff | New users | % total | Length bin | #Users | Description |
|--------|-----------|---------|------------|--------|-------------|
| Dec 2000 | 343 | 5.7% | 1–4 | 0 | removed in preprocessing |
| Jun 2001 | 6 | 0.1% | 5–9 | 8 | sparse users |
| Jan 2002 | 3 | 0.0% | 10–19 | 68 | medium-history users |
| Jul 2002 | 2 | 0.0% | 20+ | 267 | long-history users |

We therefore adopt a `by_length` split: users with 4–20 interactions form the target set and users with more than 20 form the source set, defining cold-start by short interaction history rather than temporal novelty. This yields 5,084 source users (mean length 110.3, median 74) and 951 target users (mean length 15.1, median 16) over 3,525 items. Crucially, this split introduces no temporal boundary — source and target users are active over the same period (both spanning Apr 2000 to early 2003), with highly similar first- and last-interaction timestamp distributions as illustrated in Figure 5. The ML-1M `by_length` split is therefore structurally closer to the `by_random` split on Amazon: target users are sparse users from the same time period, rather than genuinely new users from a future window. We treat ML-1M as a controlled short-history benchmark that complements, but does not replace, the stricter time-based Amazon evaluation.

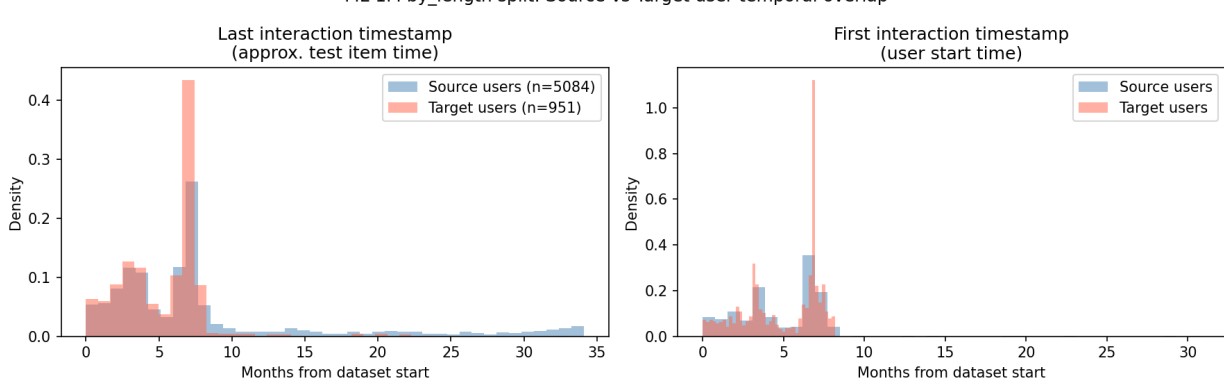

Figure 5: Temporal overlap between source and target users under the ML-1M `by_length` split. Source and target users have highly similar first- and last-interaction timestamp distributions, indicating no clear temporal boundary.

### H.2 Full-Item Ranking Results

All models are evaluated using full-item ranking without negative sampling. Table 15 reports NDCG@10, Recall@10, and MRR under `max_len = 4`. DACSR++ achieves the best NDCG@10 and MRR, while TL obtains the best Recall@10. Compared with TL, DACSR++ improves NDCG@10 and MRR, while being lower on Recall@10. CL4SRec also performs competitively, outperforming SASRec and LinRec, which

suggests that contrastive augmentation can partially compensate for explicit transfer in this short-history setting. SIGMA performs poorly, consistent with its results on the Amazon datasets.

The small performance gap among DACSR++, TL, and CL4SRec suggests that ML-1M `by_length` is less challenging than the time-based cold-start setting, since source and target users share highly overlapping temporal distributions. AMID performs substantially worse, likely because it relies on stable batch-level inter-/intra-domain alignment. In our setting, users are non-overlapping across source and target groups, and the target sequences are extremely short. Consequently, the target-side representations can be noisy, making AMID's alignment objectives less effective. Overall, ML-1M provides a complementary benchmark to the Amazon datasets. Its `by_length` split should be interpreted as a controlled short-history evaluation, similar in spirit to the `by_random` split, rather than a strict temporal cold-start scenario.

Table 15: Full-item ranking results on ML-1M under `max_len` = 4.

|  | AMID | SIGMA | SASRec | LinRec | CL4SRec | **DACSR++** | TL |
|---|---|---|---|---|---|---|---|
| NDCG@10 | 0.0552 | 0.0792 | 0.0917 | 0.1050 | 0.1086 | **0.1129** | 0.1112 |
| Recall@10 | 0.1105 | 0.1198 | 0.1451 | 0.1750 | 0.1748 | 0.1862 | **0.1892** |
| MRR | 0.0506 | 0.0783 | 0.0880 | 0.0964 | 0.1001 | **0.1036** | 0.0999 |

## I  Extension to QUILT based Target-aware Sampling

We extend the uniform random sampling probability $p$ in DACSR++ to a target-aware QUILT-inspired sampling strategy. We borrow two high-level ideas from QUILT: temporal segmentation and Bayesian optimization over segment-level policies. However, our adaptation differs from the original QUILT in a fundamental way.

QUILT partitions the data stream into temporal windows and may discard stale windows entirely, retaining only the segments considered most relevant. We found that applying this idea directly, i.e., restricting training to only a subset of source users, substantially degrades performance because it reduces the effective training data. Therefore, instead of removing source users, we retain all source users and vary how each user's history is sub-sampled when constructing simulated cold-start sequences. Specifically, source users are ranked by their last interaction timestamp and divided into $K$ temporal segments. For a user in segment $k$, each item in the source history is independently retained with probability $p_k$, producing a simulated target sequence with variable length. In this formulation, the segment policy is no longer an include/exclude decision over data windows, as in QUILT, but a segment-specific per-item keep probability. Less relevant segments can be assigned lower $p_k$, yielding shorter simulated target sequences and reducing their influence on target-side training, while more relevant segments can be assigned higher $p_k$. Setting $p_k = p$ for all segments recovers the original DACSR++ global masking strategy.

The segment number $K \in \{1, 2, 3, 5, 7\}$, the keep-probability vector $[p_0, \ldots, p_{K-1}]$, and the loss weight $\beta_{SS}$ for emulated targets are jointly tuned using Bayesian optimization on cold-start validation NDCG@10. The uniform setting $p_k = p$, the optimal for DACSR++, is used as the warm-start probe.

**Empirical results.** We evaluate the proposed QUILT-inspired sampling strategy on five Amazon datasets under both time-based and random cold-start splits. Table 16 reports NDCG@10 for DACSR++ and the QUILT-inspired variant under each split.

Under the random split, temporal segmentation consistently improves NDCG@10 across all five datasets, with gains ranging from +1.17% to +3.26%. The learned $p_k$ values are often highly non-uniform, suggesting that different temporal cohorts benefit from different sub-sampling levels when constructing simulated cold-start sequences; cohorts whose histories are less representative of sparse target users can be assigned lower $p_k$, reducing their influence on the simulated target encoder. Under the time-based split, gains are more mixed — three datasets improve while two show slight declines. This is expected because cold-start users already come from the most recent period in this setting, and the direct cold-start training objective with validation-tuned

Table 16: NDCG@10 of DACSR++ vs. QUILT-inspired sampling under time-based (top) and random (bottom) splits.

| Dataset | Ours | QUILT $K$ | QUILT $p_k$ | QUILT | $\Delta$ |
|---|---|---|---|---|---|
| *Time-based split* | | | | | |
| Beauty | 0.5967 | 5 | [0.49, 0.22, 0.80, 0.65, 0.63] | 0.6005 | +0.64% |
| Toy | 0.5763 | 2 | [0.60, 0.38] | 0.5806 | +0.75% |
| Sport | 0.5710 | 2 | [0.78, 0.75] | 0.5672 | −0.67% |
| Kindle | 0.9191 | 2 | [0.48, 0.34] | 0.9290 | +1.08% |
| Electronics | 0.4854 | 3 | [0.67, 0.75, 0.64] | 0.4850 | −0.08% |
| *Random split* | | | | | |
| Beauty | 0.4357 | 3 | [0.75, 0.40, 0.42] | 0.4430 | +1.68% |
| Toy | 0.4510 | 7 | [0.89, 0.81, 0.90, 0.46, 0.77, 0.31, 0.88] | 0.4657 | +3.26% |
| Sport | 0.4795 | 3 | [0.26, 0.81, 0.62] | 0.4881 | +1.79% |
| Kindle | 0.5540 | 1 | [0.54] | 0.5605 | +1.17% |
| Electronics | 0.4433 | 3 | [0.46, 0.87, 0.71] | 0.4566 | +3.00% |

$\beta_{SS}$ already attenuates stale source-side patterns, leaving less room for segment-level temporal re-weighting to contribute.

**Tuning cost.** The QUILT-inspired adaptation introduces additional hyperparameters, including the number of segments $K$, the segment-specific keep probabilities $p_k$, and the loss weight $\beta_{SS}$. These are jointly tuned by Bayesian optimization. Although Bayesian optimization does not guarantee a global optimum, it is more efficient than exhaustive grid search and reduces manual tuning effort. Nevertheless, its wall-clock cost remains substantial in our experiments, as we use 10 random exploration trials followed by 30 Gaussian-process-guided optimizations for each setting.

