# OpenReview forum: "Domain Adaptation for Cold-Start Users in Sequential Recommendation"
_TMLR — Accepted by TMLR_

### Review · Reviewer_PTpQ · 2026-03-15

**Summary Of Contributions:**

The paper proposes DACSR, a dual-transformer framework that tackles cold-start sequential recommendation as a domain adaptation problem by framing regular users (long histories) and cold-start users (short histories) as source and target domains respectively. The key idea is constructing an emulated target domain by subsampling source sequences, then using contrastive learning to align the emulated and original source representations. Three progressively enhanced variants are introduced: DACSR, DACSR+ (introduces popularity-weighted user representations to reduce item bias), and DACSR++ (which further adds user similarity-weighted contrastive loss). Experiments on five Amazon datasets show consistent improvements over baselines like SASRec, MetaTL, LinRec, and SIGMA. The main concerns are around the evaluation methodology, a consistent regression of DACSR++ on Kindle across both splits, and a few inconsistencies in the paper.

**Additional Comments:**

The paper addresses an important problem with a well-structured framework. The concerns above are about strengthening the evidence rather than the core approach. Addressing the evaluation methodology and the inconsistency on Kindle dataset in particular would improve overall clarity and make this a stronger submission.

**Audience:**

Yes

**Audience Explanation:**

Cold-start sequential recommendation is a well-studied and practically important problem, hence the scope of this paper is highly relevant to TMLR's audience.

**Broader Impact Concerns:**

No significant concerns as the datasets used are publicly available Amazon product reviews. One minor point worth acknowledging is that the emulated target domain is built by uniform random subsampling of source sequences, which naturally over-represents popular items. The popularity-weighted user representation in DACSR+ and DACSR++ partially addresses this issue, but the training data for the emulated domain remains biased toward popular items. In deployment this could reinforce exposure to popular items for cold-start users instead of enabling discovery.

**Claims And Evidence:**

No

**Claims Explanation:**

The paper's results are broadly consistent with its claims, but there are a few issues that need to be addressed:

The evaluation follows Wang et al. (2021) in randomly sampling 100 negative items per ground truth for ranking. However, Krichene & Rendle (KDD 2020) showed that metrics computed over sampled candidate sets can disagree with full-ranking evaluation. This applies to both the random and time-based splits and is relevant here because some reported improvements are quite small, for instance, +0.15% on Electronics NDCG@10 at max_len=20 (Table 6) and +0.84% on Kindle Hit@10 (Table 3). Supplementing with full-ranking evaluation on at least one dataset would make the claims more convincing.

Section 3.6 states that DACSR++ achieves the best performance and is used as the primary model. However, Tables 5/7 shows that on Kindle, DACSR+ outperforms DACSR++ under both the random split (DACSR++ NDCG@10 = 0.5540 vs. DACSR+ = 0.5758) and the time-based split (DACSR++ = 0.9191 vs. DACSR+ = 0.9320). This is a consistent pattern on Kindle dataset, and it directly contradicts the claim in Section 3.6. The authors should explain what characteristics of the Kindle dataset led to this behavior, or adjust the claim to reflect that DACSR++ is generally but not always the best variant.

The random split evaluates cold-start by converting regular users who originally had at least 10 interactions down to max_len=4. This is a valid length-shift experiment but is not the same as evaluating true cold-start users. The time-based split, where target users have as few as 4 real interactions, is the more credible cold-start evaluation. The paper does not clearly distinguish between these two settings, and the cold-start claims in the introduction and conclusion read as more general than the experiments strictly support.

Finally, max_len=4 is used as the primary evaluation setting throughout but the choice is not explicitly justified. Table 6 explores max_len ∈ {4, 6, 10, 20} but only compares DACSR++ against SASRec, providing no evidence that the gains over MetaTL, LinRec, and SIGMA hold at longer sequence lengths.

**Requested Changes:**

The most important concern is the evaluation methodology. Both splits (random/time based) use 100 sampled negatives per ground truth, and some of the reported improvements are small enough that it is hard to draw firm conclusions under this setting. Full-ranking evaluation, or results with a larger sample, on at least a subset of datasets would substantially strengthen the paper's claims.

The regression of DACSR++ on Kindle needs to be addressed. As noted above, DACSR+ consistently outperforms DACSR++ on Kindle under both splits in Tables 5 and 7. Since Section 3.6 makes a categorical claim about DACSR++ being the best variant, the authors should either explain this pattern or revise the claim.

The paper would benefit from being clearer about what the random split actually evaluates. Since target users in the random split originally had at least 10 interactions and are artificially sampled, this is a controlled length-shift experiment rather than a true cold-start evaluation. Stating this explicitly would make the paper's scope clearer and position the time-based results as the primary cold-start evidence.

The choice of max_len=4 as the headline setting should be briefly justified, and Table 6 should include all the baselines rather than just SASRec, so readers can assess whether the performance gap narrows uniformly across all methods as sequences get longer.
A few smaller issues: the caption of Figure 3 on page 17 says "improved separation indicates more effective domain adaptation," which contradicts Figure 2's caption and the paper's stated goal of domain mixing and should be corrected. The terms "compound shift" and "complex shift" and "compounded shift" are used interchangeably across the abstract, page 2, and Section 3.1.2 and should be standardized to one term throughout. There is also a double period at the end of page 18, a typo in the spelling of "hyperparameter" on page 8, and a mismatched parenthesis in Eq. (9).

---

> ### Author Response · Authors · 2026-04-03
> **Responses to the comments from Reviewer PTpQ**
>
> We thank the reviewer for these comments. In the revision, we will strengthen the evaluation discussion by adding explicit full-ranking as additional evidence, revise the framing of the random split as a controlled length-shift setting, soften the categorical claim about DACSR++ to account for the Kindle exception, expand the `max_target_len` comparison table to include multiple baselines, and correct the terminology/caption/typo noted.
>
> **Reviewer concern:**
>  100-sampled negatives per ground-truth interaction is used for evaluation. The reviewer asks for full-ranking evaluation due to the concern that may have different conclusions with 100-negative evaluation.
>
> **Response:**
> We agree that sampled-negative evaluation may not be consistent to the full-ranking results. To address this concern, we will present additional full-ranking results to strengthen the main claims.
> We have completed full-ranking reevaluation of the saved best checkpoints on **all five datasets** for four models for time-split experiments:  `DACSR++`,  `SIGMA`, `LinRec` and `TL` shown in table below, where the best result is in bold, the second-best is indicated by (2nd), and `Impr.` is the percentage difference between `DACSR++` and the strongest competing baseline in that row. A negative `Impr.` means `DACSR++` is below the best baseline.
>
> | Dataset | Metric | DACSR++ | SIGMA | LinRec | TL | Impr. |
> | --- | --- | ---: | ---: | ---: | ---: | ---: |
> | Beauty | Hit@1 | **0.0465** | 0.0395 (2nd) | 0.0371 | 0.0200 | +17.72% |
> | Beauty | Hit@5 | **0.1489** | 0.1329 | 0.1331 (2nd) | 0.1146 | +11.87% |
> | Beauty | Hit@10 | **0.2106** | 0.2028 (2nd) | 0.1883 | 0.1863 | +3.85% |
> | Beauty | NDCG@5 | **0.1005** | 0.0874 (2nd) | 0.0867 | 0.0682 | +14.99% |
> | Beauty | NDCG@10 | **0.1201** | 0.1099 (2nd) | 0.1044 | 0.0917 | +9.28% |
> | Beauty | MRR | **0.1032** | 0.0901 (2nd) | 0.0880 | 0.0723 | +14.54% |
> | Toy | Hit@1 | 0.0613 | 0.0599 | **0.0768** | 0.0632 (2nd) | -20.18% |
> | Toy | Hit@5 | 0.1294 (2nd) | 0.1238 | **0.1415** | 0.1138 | -8.55% |
> | Toy | Hit@10 | 0.1634 (2nd) | 0.1465 | **0.1727** | 0.1509 | -5.39% |
> | Toy | NDCG@5 | 0.0962 (2nd) | 0.0948 | **0.1102** | 0.0913 | -12.70% |
> | Toy | NDCG@10 | 0.1073 (2nd) | 0.1022 | **0.1205** | 0.1034 | -10.95% |
> | Toy | MRR | 0.0976 (2nd) | 0.0930 | **0.1107** | 0.0952 | -11.83% |
> | Sports | Hit@1 | **0.0241** | 0.0195 | 0.0203 (2nd) | 0.0189 | +18.72% |
> | Sports | Hit@5 | **0.0684** | 0.0535 | 0.0599 (2nd) | 0.0586 | +14.19% |
> | Sports | Hit@10 | **0.1017** | 0.0730 | 0.0973 (2nd) | 0.0876 | +4.52% |
> | Sports | NDCG@5 | **0.0478** | 0.0366 | 0.0405 (2nd) | 0.0395 | +18.02% |
> | Sports | NDCG@10 | **0.0587** | 0.0427 | 0.0524 (2nd) | 0.0486 | +12.02% |
> | Sports | MRR | **0.0535** | 0.0387 | 0.0454 (2nd) | 0.0438 | +17.84% |
> | Kindle | Hit@1 | **0.1635** | 0.0965 | 0.1445 | 0.1464 (2nd) | +11.68% |
> | Kindle | Hit@5 | **0.3583** | 0.2181 | 0.2739 | 0.2922 (2nd) | +22.62% |
> | Kindle | Hit@10 | **0.4240** | 0.2819 | 0.3515 | 0.3818 (2nd) | +11.05% |
> | Kindle | NDCG@5 | **0.2630** | 0.1603 | 0.2115 | 0.2179 (2nd) | +20.70% |
> | Kindle | NDCG@10 | **0.2843** | 0.1807 | 0.2362 | 0.2458 (2nd) | +15.66% |
> | Kindle | MRR | **0.2546** | 0.1641 | 0.2132 | 0.2189 (2nd) | +16.31% |
> | Electr. | Hit@1 | 0.0126 (2nd) | 0.0083 | 0.0124 | **0.0136** | -7.35% |
> | Electr. | Hit@5 | **0.0562** | 0.0283 | 0.0453 | 0.0473 (2nd) | +18.82% |
> | Electr. | Hit@10 | **0.0820** | 0.0471 | 0.0683 | 0.0751 (2nd) | +9.19% |
> | Electr. | NDCG@5 | **0.0340** | 0.0184 | 0.0287 | 0.0304 (2nd) | +11.84% |
> | Electr. | NDCG@10 | **0.0424** | 0.0244 | 0.0362 | 0.0393 (2nd) | +7.89% |
> | Electr. | MRR | **0.0369** | 0.0226 | 0.0323 | 0.0348 (2nd) | +6.03% |
>
> These full-ranking results strengthen two points. First, the results are generally consistent with those of the 100-negative setting. Second, the conclusions under full rank is also general consistent to that of 100-negative evaluation: `DACSR++` remains strong overall, but not uniformly best on every dataset (for example, Toy dataset).
>
> As a concrete train-and-test confirmation beyond reevaluating old checkpoints, we also performed new full-ranking training on Toy. After training using full-rank validation and tuning hyperparameter of the models, the current best Toy `DACSR++` and `LinRec` full-evaluation results can be compared as follows:
>
> | Model | Recall@1 | Recall@5 | Recall@10 | NDCG@1 | NDCG@5 | NDCG@10 | MRR |
> | --- | ---: | ---: | ---: | ---: | ---: | ---: | ---: |
> | DACSR++ | 0.0924 | 0.1695 | 0.2046 | 0.0924 | 0.1316 | 0.1430 | 0.1319 |
> | LinRec | 0.0774 | 0.1609 | 0.1975 | 0.0774 | 0.1228 | 0.1348 | 0.1218 |
> | Impr. | +19.31% | +5.34% | +3.56% | +19.31% | +7.13% | +6.06% | +8.24% |
>
> In the revision, we will include full-ranking, time-split evaluations as additional evidence and present full-ranking Toy retraining results as a more rigorous train–test verification, especially for the Toy-case exception.

---

> ### Author Response · Authors · 2026-04-03
> **Response to comments from Reviewer PTpQ**
>
> **Reviewer concern:**
> DACSR+ consistently outperforms DACSR++ on Kindle, so the categorical claim that DACSR++ is the best variant should be revised or explained.
>
> **Response:**
> We agree that the original categorical wording is too strong. In the revision, we will revise the claim from “DACSR++ is the best variant” to a more accurate statement: DACSR++ is generally the strongest variant overall, but Kindle is an exception.
>
> We further investigate user similarity patterns in the Kindle dataset, as DACSR++ explicitly leverages additional user similarity signals. Our analysis indicates that the source-domain similarity signal in Kindle is significantly weaker and less informative compared to other datasets.
> To quantify this, we compute pairwise user cosine similarity based on learned user embeddings and examine the similarity distributions in the source domain. The distributions for different datasets will be included in the revised paper for direct comparison. The corresponding summary statistics of the similar between users are as follows:
>
> | Dataset | Mean | Std | Median |
> | --- | ---: | ---: | ---: |
> | Beauty | 0.1726 | 0.2427 | 0.1121 |
> | Toy | 0.1479 | 0.2405 | 0.0977 |
> | Sport | 0.1519 | 0.2672 | 0.0999 |
> | Kindle | 0.0737 | 0.2530 | 0.0389 |
> | Electronics | 0.1697 | 0.2690 | 0.1261 |
>
> So Kindle is of the clear low-similarity comparing to other datasets. The whole similarity distribution is shifted lower, and the figures show a heavier negative side as well (in the revised paper). This clue that the user similarity structure in source domain is much weaker on Kindle, might make the similarity-guided transfer in DACSR++ less informative there.
>
> We will add in the revised paper:
>
> Kindle appears to be a harder case because the source-domain similarity structure is substantially weaker than in the other datasets, so the similarity-guided transfer in DACSR++ might be less beneficial than the simpler DACSR+ variant.
>
> Thus, in the revision we will both:
>
> 1. revise the categorical claim, and
> 2. add this Kindle-specific explanation supported by the new user-similarity analysis.

---

> ### Author Response · Authors · 2026-04-03
> **Responses to the comments from Reviewer PTpQ**
>
> **Reviewer concern:**
> The random split is not a true cold-start evaluation, because target users originally had at least 10 interactions and are then artificially sampled. The reviewer suggests describing it as a controlled length-shift setup and treating time-based split as the primary cold-start evidence.
>
> **Response:**
> We agree with this clarification and will revise the paper accordingly.
>
> In the revision, we will explicitly state that:
>
> - the **random split** should be interpreted as a **controlled target-length reduction / length-shift experiment**, not as a natural cold-start scenario, and
> - the **time-based split** is the primary evidence for realistic cold-start transfer.
>
> This revision will make the scope of the two settings clearer:
>
> - random split: controlled analysis of shortened target-history regimes,
> - time-based split: primary cold-start evaluation.
>
> We believe this revision enhances the clarity of the presentation while leaving the underlying results unchanged. We thank the reviewer for the helpful feedback.

---

> ### Author Response · Authors · 2026-04-03
> **Responses to the comments from Reviewer PTpQ**
>
> **Reviewer concern:**
> The choice of `max_len=4` should be justified, and Table 6 should include all baselines rather than only SASRec.
>
> **Response:**
> We agree. The main reason `max_target_len=4` is emphasized is that it corresponds to the **coldest / shortest-target-history** setting in our study, where transfer should matter most. However, we agree that this rationale should be stated explicitly rather than assumed.
>
> We will therefore revise the text to explain that: `max_target_len=4` is used as the headline setting because it represents the most severe target-history constraint in our benchmark.
>
> We also agree that Table 6 should include all baselines. We already compiled the broader `max_target_len` comparisons across `TL`, `SIGMA`, and `LinRec`. In the revision, we will expand the corresponding table so readers can see whether the gap narrows consistently across all methods rather than only relative to SASRec.
>
> For clarity, below is the updated random-split `NDCG@10` summary with corrected `Impr.` values (computed as `%` of `DACSR++` against the strongest competing baseline in each dataset column). The best is bold and second best is in Italic.
>
> | max_len | Model   | Electr     | Impr.  | Kindle     | Impr.  | Beauty     | Impr.  | Toy        | Impr.  | Sports     | Impr.  |
> | ------- | ------- | ---------- | ------ | ---------- | ------ | ---------- | ------ | ---------- | ------ | ---------- | ------ |
> | 20      | DACSR++ | *0.4824*   | -0.33% | *0.6272*   | -0.06% | **0.4368** | 16.08% | **0.4214** | 11.54% | **0.4373** | 22.12% |
> |         | SASRec  | 0.4817     |        | **0.6276** |        | *0.3763*   |        | *0.3778*   |        | *0.3581*   |        |
> |         | TL      | **0.4840** |        | 0.6151     |        | 0.3748     |        | 0.3567     |        | 0.3524     |        |
> |         | SIGMA   | 0.4246     |        | 0.4104     |        | 0.3680     |        | 0.3005     |        | 0.2911     |        |
> |         | LinRec  | 0.4685     |        | 0.6060     |        | 0.3744     |        | 0.3549     |        | 0.3524     |        |
> | 10      | DACSR++ | **0.4811** | 6.30%  | **0.6360** | 1.35%  | **0.4566** | 21.47% | **0.4434** | 27.08% | **0.4717** | 25.82% |
> |         | SASRec  | 0.4510     |        | *0.6275*   |        | 0.3720     |        | 0.3468     |        | 0.3744     |        |
> |         | TL      | *0.4526*   |        | 0.6132     |        | 0.3675     |        | *0.3489*   |        | *0.3749*   |        |
> |         | SIGMA   | 0.4031     |        | 0.3450     |        | 0.3738     |        | 0.2854     |        | 0.3269     |        |
> |         | LinRec  | 0.4433     |        | 0.5811     |        | *0.3759*   |        | 0.3482     |        | 0.3595     |        |
> | 6       | DACSR++ | **0.4879** | 10.71% | **0.6173** | 2.76%  | **0.4517** | 19.76% | **0.4314** | 28.97% | **0.4847** | 17.42% |
> |         | SASRec  | 0.4241     |        | 0.5808     |        | 0.3714     |        | 0.3236     |        | 0.3833     |        |
> |         | TL      | 0.4016     |        | *0.6007*   |        | 0.3747     |        | 0.2875     |        | *0.4128*   |        |
> |         | SIGMA   | 0.4117     |        | 0.3651     |        | 0.3751     |        | 0.2887     |        | 0.3622     |        |
> |         | LinRec  | *0.4407*   |        | 0.5812     |        | *0.3772*   |        | *0.3345*   |        | 0.3739     |        |
> | 4       | DACSR++ | **0.4433** | 9.08%  | **0.5540** | 0.67%  | **0.4357** | 12.15% | **0.4510** | 28.53% | **0.4795** | 17.52% |
> |         | SASRec  | 0.3871     |        | 0.5430     |        | 0.3450     |        | 0.3294     |        | 0.3911     |        |
> |         | TL      | *0.4064*   |        | *0.5503*   |        | *0.3885*   |        | *0.3509*   |        | *0.4080*   |        |
> |         | SIGMA   | 0.3574     |        | 0.3730     |        | 0.3616     |        | 0.2901     |        | 0.3642     |        |
> |         | LinRec  | 0.3965     |        | 0.5253     |        | 0.3564     |        | 0.3246     |        | 0.3816     |        |
>
>
> We will also soften the interpretation slightly: the performance gap does tend to narrow as the target sequence becomes longer, but the extent of narrowing is not perfectly uniform across every model/dataset pair.

---

> ### Author Response · Authors · 2026-04-03
> **Responses to the comments from Reviewer PTpQ**
>
> **Reviewer concern:**
> The Figure 3 caption says “improved separation indicates more effective domain adaptation,” which contradicts Figure 2 and the paper’s stated goal of domain mixing.
>
> **Response:**
> We will correct this. The intended interpretation is that better adaptation should correspond to **more meaningful source-target alignment / mixing**, not stronger separation in the opposite sense. We will revise the Figure 3 caption so it is consistent with Figure 2 and with the paper’s main explanation of domain adaptation.
>
>
> **Reviewer concern:**
> The terms are used inconsistently.
>
> **Response:**
> We agree and will standardize the terminology throughout the paper. We currently plan to use a single term consistently:
>
> - `compound shift`
>
> and replace the other variants accordingly.
>
>
> **Reviewer concern:**
> The reviewer points out:
>
> - a double period on page 18,
> - a typo in “hyperparameter” on page 8,
> - and a mismatched parenthesis in Eq. (9).
>
> **Response:**
> We thank the reviewer for catching these issues. We will fix all of them in the revised manuscript.

---

> > ### Comment · Reviewer_PTpQ · 2026-04-14
> > **Thanks for the updates**
> >
> > The additional analysis has addressed all the issues and enhanced the paper's quality.

---

> > > ### Author Response · Authors · 2026-04-21
> > > **Thanks for the comments**
> > >
> > > We thank the reviewer for the helpful comments that have improved the quality of this work. All discussed results and revisions will be incorporated into the updated manuscript.

---

### Review · Reviewer_4dmv · 2026-04-02

**Summary Of Contributions:**

This paper studies the cold-start recommendation problem by framing it as a domain adaptation task, where users with rich interaction histories (source domain) are used to improve recommendations for users with minimal histories (target domain). The paper proposes a transfer learning framework that aligns representations between these two groups. Empirically, the method is evaluated on five datasets derived from Amazon product reviews, showing improvements over several baseline recommendation models.

Strengths

1. Studies an important and long-standing problem in recommendation systems (cold-start).
2. Attempts to unify cold-start under a domain adaptation perspective.
3. Empirical improvements over selected baselines on multiple datasets.
4. The formulation is conceptually simple and potentially generalizable.

Weaknesses

1. Contribution relative to decades of cold-start and transfer learning literature is unclear.
2. The “domain adaptation” formulation (long-history vs short-history users) is questionable and insufficiently justified.
3. Baselines appear incomplete relative to prior work in cold-start recommendation.
4. Dataset diversity is limited (all Amazon review subsets), raising concerns about generalization.
5. Evaluation protocol concerns.

**Audience:**

Yes

**Audience Explanation:**

Cold-start recommendation remains a fundamental challenge in recommender systems

**Broader Impact Concerns:**

No major ethical concerns beyond standard recommender system issues

**Claims And Evidence:**

No

**Claims Explanation:**

While the empirical results suggest improvements over the included baselines, several issues limit how convincing the evidence is.

1. Missing or insufficiently discussed related work: Cold-start recommendation has been studied extensively for decades, including content-based and hybrid methods, meta-learning / few-shot recommendation, cross-domain recommendation, representation learning for sparse users, user modeling under sparsity.

The current paper does not clearly position itself relative to these directions. In particular, the idea of transferring knowledge from dense users to sparse users is not new, and has appeared in multiple forms (e.g., knowledge distillation, meta-learning, shared embeddings).

Key questions:
Can the authors provide a comprehensive comparison with prior cold-start methods, especially recent deep learning approaches (e.g., meta-learning, contrastive learning, and hybrid models)? What is the precise contribution beyond reframing as “domain adaptation”?

2. Questionable domain adaptation formulation: The core assumption is that, users with long histories = source domain, users with short histories = target domain. This raises conceptual concerns: These are not clearly distinct domains (same platform, same items, same interaction distribution). The difference is primarily data density, not distribution shift. This makes the formulation somewhat artificial.

Key questions:
What is the empirical evidence that long-history and short-history users exhibit domain shift rather than just sparsity?
Why is domain adaptation the right abstraction instead of, e.g., regularization, imputation, or uncertainty-aware modeling?
Do the authors measure distribution divergence between these “domains”?

3. Baseline selection is insufficient: The baselines seem limited relative to the literature. In particular, missing comparisons may include:
Strong cold-start baselines (content-based, hybrid, or meta-learning); Recent graph-based recommenders with sparse user handling; Contrastive/self-supervised methods for representation robustness, etc.

If the main comparison is against standard collaborative filtering variants, improvements may not be meaningful.

Key question:
Can the authors include stronger and more recent cold-start baselines, or justify why the current baselines are sufficient?

4. Dataset limitations (all Amazon subsets): The paper evaluates on five datasets, but all are derived from Amazon product reviews.

This raises concerns: These datasets are highly correlated (same platform, similar structure). Results may not generalize well. Domain adaptation claims are harder to validate without true cross-domain variation. Many works complement with datasets like MovieLens, Yelp, or industrial logs.

Key questions:

Why are all datasets from the same source? Can at least one non-Amazon dataset be included?

5. Evaluation protocol concerns. The paper constructs the cold-start setting partly via an “emulated target domain” obtained by sampling short subsequences from long user histories. While this provides a controlled way to study length shift, it differs from realistic cold-start scenarios where users are genuinely new and may exhibit different preference distributions.

How well does this setup capture the challenges of real-world cold-start recommendation, beyond sequence-length sparsity?

**Requested Changes:**

1. Strengthen related work and positioning

2. Justify the domain adaptation formulation

3. Add stronger baselines

4. Improve dataset diversity

5. Clarify evaluation protocol

---

> ### Author Response · Authors · 2026-04-21
> **Response to reviewer 4dmv on comment of "Strengthen related work and positioning"**
>
> **Reviewer concern:**
> The reviewer asks for a comprehensive related work，especially recent approaches including meta-learning, contrastive learning, and hybrid models，and a clearer positioning on the technical contribution beyond reframing the problem as domain adaptation.
>
> **Response — Part 1 of 3:** Changes to the Related Work
>
> The current Related Work discusses source-assisted transfer and motivates the interaction-only setting, but it does not sufficiently organize the literature into method families, under-emphasizes relevant categories (contrastive sequential recommendation and cross-domain sequential recommendation), and does not clearly state what DACSR++ adds on top of a domain-adaptation framing. We have rewritten §Related Work, added an explicit scope constraint and a positioning paragraph.
>
> **Opening paragraph.** The revised §Related Work opens with a one-paragraph scope statement: interaction sequences only, no user/item features, no knowledge graphs, and no user- or item-overlap assumption between source and target populations. This frames why side-information and shared-user methods are *discussed* but not used as baselines.
>
> **§2.1 Sequential Recommendation.** We shorten the traditional CF/MF background (KNN, MF, BPR, SLIM, NeuralMF, VAE-CF, DIN), since those are not the main baselines in this work. We keep the Markov-chain → CNN → RNN → transformer evolution ending at SASRec/BERT4Rec/TiSASRec, and **add two recent architectural variants**: LinRec (linear attention) and SIGMA (selective-gated Mamba / state-space backbone). The paragraph ends by labeling this whole family an *implicit* knowledge-transfer baseline when trained on combined source+target data. The contrastive-learning paragraph (CL4SRec, S3-Rec, etc.,) is retained but ends with the clarification that these methods apply the same training objective to all users regardless of history length and therefore provide no mechanism to align representations across the sequence-length gap.
>
> **§2.2 Cold-Start Recommendation.** We keep the two-category split (side information vs. model generalization) but prune the side-information subsection and mark it out-of-scope rather than listing many papers we cannot compare against. In the generalization subsection, we explicitly call out MetaTL as the interaction-only sequential cold-start meta-learning representative (MeLU/MAMO remain cited but noted as attribute-requiring).
>
> **§2.3 Cross-Domain Sequential Recommendation (CDSR) (NEW).** The full proposed text:
>
> > **§2.3 Cross-Domain Sequential Recommendation.** Cross-domain sequential recommendation (CDSR) [Zhu] traditionally bridges two domains via *shared/overlapping users*. EMCDR [EMCDR], Chen et al. [Chen], and CDRIB [CDRIB] learn a mapping between user representations across domains and therefore require a substantial shared-user base. More recently, AMID [AMID] relaxes this to an *open-world* assumption in which only partial user overlap is available: it constructs interest groups via inner- and inter-domain compositions that accommodate both overlapping and non-overlapping users, and debiases the observed cross-domain signal. CDSR methods that further require item-side knowledge graphs or content features — e.g., HeroGraph[HeroGraph], MIFN [MIFN] and BiTGCF [BiTGCF] — are out of scope in our benchmark.
>
> **§2.4 Positioning of DACSR++ (NEW).** Full proposed text:
>
> > **§2.4 Positioning of DACSR++.** Within the applicable scope, the directly comparable baselines fall into five groups: (i) implicit transfer from combined source and target (SASRec, LinRec, SIGMA), (ii) explicit source-assisted transfer (TL), (iii) interaction-only meta-learning (MetaTL), (iv) contrastive self-supervised sequential methods (CL4SRec), and (v) open-world CDSR (AMID).
> >
> > DACSR++ differs from all five in a structural way. (a) Unlike (i)–(ii), it does not treat the source only as extra training data or as a pretraining; it constructs a *simulated target domain* — short subsequences of source users — to provide labelled examples in the cold-start length regime during training. (b) Unlike (iii), it does not rely on episode-level task construction but imposes a continuous similarity-weighted alignment objective over the source side. (c) Unlike (iv), its contrastive signal is not within-sequence augmentation robustness but cross-view alignment between a user's full-length and short-length representations. (d) Unlike (v), it does not use similarity as a symmetric peer-selection gate across domains — which is unstable when the target side has few interactions — but as a *unidirectional, source-side* optimization signal that regularizes the representation space.
> >
> > The contribution is therefore not the relabelling of cold-start as domain adaptation, but the three concrete mechanisms that realize that framing: a simulated-target training bridge, a source-side similarity-weighted contrastive objective, and explicit length-invariant same-user alignment.

---

> ### Author Response · Authors · 2026-04-21
> **Response to reviewer 4dmv on comment of "Strengthen related work and positioning"**
>
> ### Response -- Part 2 of 3: Applicable-Baseline Category Table
>
> The new §2.4 positioning subsection is backed by the following categorization. The key point is not only which methods exist, but which are actually applicable under our scope:
>
> | Category | Representative work | Requires side info? | Applicable to our setting | Why included / excluded |
> | --- | --- | --- | --- | --- |
> | **Meta-learning** | MAML, MeLU, MAMO, MetaTL | Some (MeLU/MAMO need attributes) | Partially | MetaTL (interaction-only, temporal) **included**; MAMO/MeLU excluded — need user/item attributes |
> | **Contrastive self-supervised** | CL4SRec, DuoRec, CLCR | CLCR | Yes | CL4SRec **newly added** in revision |
> | **Implicit source-assisted transfer (pooled training)** | SASRec, LinRec, SIGMA | No | Yes | Trained on combined source+target data with no explicit source/target separation. LinRec (SIGIR 2023) and SIGMA (AAAI 2025) are recent architectural variants — linear attention and selective-gated Mamba respectively |
> | **Explicit source-assisted transfer** | TL | No | Yes | Uses explicit transfer — source pretraining followed by target fine-tuning |
> | **Cross-domain sequential (user-overlap based)** | EMCDR, AMID, CDRIB | EMCDR / CDRIB need full user overlap | Partially | AMID **newly added** (open-world, no strict restriction on user overlap); EMCDR and CDRIB excluded — both require a full shared-user bridge between the two domains |
> | **Graph / content cold-start** | HeroGraph, MIFN, BiTGCF | Yes (item/knowledge graph) | No | All require features or relation graphs |
>
> The bottom rows are out of scope under our interaction-only assumption. Within scope, the revised baseline set therefore covers: MetaTL (meta-learning), CL4SRec (contrastive), SASRec/LinRec/SIGMA (implicit using combined source and target, with LinRec and SIGMA as recent architectural variants), TL (explicit transfer), and AMID (open-world CDSR).
>
> **References added:**
>
> - **EMCDR** — Tong Man, Huawei Shen, Xiaolong Jin, Xueqi Cheng. "Cross-Domain Recommendation: An Embedding and Mapping Approach". IJCAI 2017.
> - **Chen** — Xu Chen, Ya Zhang, Ivor W. Tsang, Yuangang Pan, and Jingchao Su. "Toward Equivalent Transformation of User Preferences in Cross Domain Recommendation." ACM Trans. Inf. Syst. 2023.
> - **Zhu** — Feng Zhu, Yan Wang, Chaochao Chen, Jun Zhou, Longfei Li, and Guanfeng Liu. "Cross-Domain Recommendation: Challenges, Progress, and Prospects." IJCAI 2021.
> - **CDRIB** — Jiangxia Cao, Jiawei Sheng, Xin Cong, Tingwen Liu, and Bin Wang. "Cross-Domain Recommendation to Cold-Start Users via Variational Information Bottleneck." ICDE 2022.
> - **AMID** — Wujiang Xu, Qitian Wu, Runzhong Wang, Mingming Ha, Qiongxu Ma, Linxun Chen, Bing Han, and Junchi Yan. "Rethinking Cross-Domain Sequential Recommendation under Open-World Assumptions." WWW 2024.
> - **BiTGCF** — Meng Liu, Jianjun Li, Guohui Li, and Peng Pan. "Cross Domain Recommendation via Bi-directional Transfer Graph Collaborative Filtering Networks." CIKM 2020.
> - **MIFN** — Muyang Ma, Pengjie Ren, Zhumin Chen, Zhaochun Ren, Lifan Zhao, Jun Ma, and Maarten de Rijke. "Exploring Mixed Information Flow for Cross-domain Sequential Recommendations." ACM Trans. Knowl. Discov. Data 16(4), 2022.
> - **HeroGraph** —  Tendai Mukande. "Heterogeneous Graph Representation Learning for multi-target Cross-Domain Recommendation." In Proceedings of the 16th ACM Conference on Recommender Systems (RecSys '22).

---

> ### Author Response · Authors · 2026-04-21
> **Response to reviewer 4dmv on comment of "Strengthen related work and positioning"**
>
> ### Response — Part 3 of 3: Precise Technical Contributions Summary Beyond "Domain Adaptation Reframing"
>
> The reviewer's underlying concern is that framing cold-start as domain adaptation may only be a relabelling. The new section 2.3 makes three concrete mechanisms explicit, each of which goes beyond what any in-scope prior method provides:
>
> #### Contribution 1: Simulated Target Domain as a Structured Training Bridge
>
> SASRec, LinRec, and SIGMA learn from source/target interactions without constructing a controlled short-sequence proxy of source users. TL transfers via source-pretrain then target-finetune without bridging the long/short regime during training. Meta-learning methods (MetaTL) build episodes from source users but do not materialize a continuous short-sequence proxy aligned with identity and similarity labels.
>
> DACSR++ introduces a **simulated target domain**: short subsequences sampled from source users' histories that serve as a training-time proxy for cold-start users. This yields labelled short-length examples that carry source-side identity and similarity labels — labels that are unavailable from real cold-start users at training time and cannot be produced by generic augmentations (e.g., CL4SRec's crop/mask/reorder).
>
> #### Contribution 2: Source-Side Similarity-Weighted Contrastive Objective
>
> Standard contrastive sequential methods (CL4SRec) form positive pairs from augmented views of the same sequence. They improve representation robustness but treat all users symmetrically — there is no notion of *which* users should be closer to *which*.
>
> DACSR++ computes pairwise similarity between source users from their full-length representations and uses this similarity to weight the contrastive objective applied over the simulated target. Users similar in the source regime are pulled together in the short-sequence regime. This is a **unidirectional source→target similarity transfer** that:
> - is not present in CL4SRec (no inter-user similarity signal),
> - differs structurally from AMID's symmetric cross-domain interest grouping (which degrades when the target side is the noisy few items regime).
>
> #### Contribution 3: Length-Invariant Alignment of the Same User's Representations
>
> No in-scope baseline explicitly enforces that a single user's full-length and truncated representations should coincide. SASRec/LinRec/SIGMA rely on data exposure. TL moves a model from long to short via sequential stages, but does not impose same-user full/short alignment. CL4SRec augments *within* a single sequence length regime but does not cross the long/short boundary.
>
> DACSR++ directly minimises the distance between a source user's full-length and simulated short-length embeddings during joint training, while similar-user pairs are pushed together. This produces a length-invariant-per-user encoder that transfers to real cold-start users at inference time.

---

> ### Author Response · Authors · 2026-04-21
> **Response to comment "Domain Adaptation Formulation"**
>
> **Reviewer concern:**
> The core assumption: long-history users = source domain, short-history users = target domain, is questioned on the grounds that these are not clearly distinct domains. The difference is primarily data density, not distribution shift. (1) domain shift vs. sparsity? (2) Why is domain adaptation the right abstraction rather than regularization, imputation, or uncertainty-aware modeling? (3) measure distribution divergence between domains?
>
> **Response:**
>
> We appreciate this challenge. We address it in four parts: (i) clarifying the role of the emulated target, (ii) a concession for the random split, (iii) justification of the timestamp split as true shift, and (iv) empirical evidence.
>
> ### Clarification: Emulated Target Is a Training Bridge, Not the Evaluation Target
>
> The **evaluation target** is not the emulated target.  Under `by_timestamp`, these are users appearing only after a temporal cutoff T, unseen for source model. All reported metrics are computed on these users.
>
> The **emulated target** — short subsequences sampled from source users' histories, is used only during **training** to model length shift. It serves three purposes: (1) improving robustness to short sequences, (2) enabling similarity-based alignment before real target users are available, and (3) supporting popularity-debiased representation learning. This emulated target is discarded at inference. Evaluation is strictly on new cold-start users, instead of the emulated one.
>
> ### Random Split -> Sparsity
>
> We agree that the **by_random split** mainly reflects sparsity. It is therefore a controlled length-shift setting where target users are truncated versions of the same population. The difference is more on data density, not distribution shift.
>
> ### Timestamp Split -> Distribution Shift
>
> The **by_timestamp split** introduces genuine shift. Source and target are separated by a temporal cutoff T:
>
> | Group | Interaction period | Item set | User cohort |
> | --- | --- | --- | --- |
> | Source | Before T | Items popular up to T | Long-history users |
> | Target | After T | Emerging items | New users |
>
> This induces shift in (i) item distribution, (ii) user cohort, and (iii) temporal context. We therefore treat this as a domain adaptation setting.
>
> ### Not Regularization or Imputation?
>
> Alternatives are closely related to our framing but are insufficient in our setting: regularization reduces overfitting but does not leverage the rich histories of source users; imputation assumes missing interactions follow the target marginal, which is biased and impoverished under temporal shift; in contrast, domain adaptation explicitly transfers knowledge from a data-rich source to a shifted, data-poor target, where source users provide a meaningful distributional reference, which in principle matches our problem.
>
> ### Empirical Evidence via Split Comparison
>
> Our two-split design can be viewed as a controlled comparison between sparsity only (random split) and sparsity with temporal distribution shift (time split). If both splits reflected the same underlying difficulty, we would expect similar method behavior and performance gaps. However, the empirical results show consistent and systematic differences between the two splits.
>
> Under the random split in Table 2 in the manuscript, TL is consistently the second-best method, indicating that explicit transfer is well-suited when the primary challenge is limited user history. In contrast, under the time-based split in Table 3 (original manuscript), the ranking changes — SASRec surpasses TL in some datasets, and the performance gaps between methods are consistently reduced. These consistent shifts in method behaviour and compressed performance gaps suggest that the time-based split introduces an additional challenge beyond sparsity, suggesting that the two splits capture fundamentally different scenarios, with the time-based split reflecting temporal distribution shift rather than merely reduced data density.
>
> ### Measuring Distribution Divergence
>
> We provide two complementary forms of evidence:
>
> - **Data-level (Table 1):** Jaccard overlap between source and target item sets. Lower overlap under `by_timestamp` indicates stronger item-distribution shift. This is a direct statistic on the split itself, independent of the model.
> - **Representation-level (Figs. 2–3):** t-SNE visualizations of learned source and target sequence embeddings. Under SASRec, the two populations occupy visibly separated regions. DACSR+ and DACSR++ progressively reduce this gap. The t-SNE plots of DACSR+ in the second columns clearly show that alignment is visibly harder under the timestamp split than the random split.
>
> Together, these suggest that the timestamp split introduces distribution shift at both the data level and the representation level.

---

> ### Author Response · Authors · 2026-04-21
> **Response to the Baseline issue from Reviewer 4dmv**
>
> **Reviewer concern:**
> The paper's baseline set is too narrow. In particular, the reviewer asks why no contrastive self-supervised or cross-domain methods are included, and whether the improvements survive comparison to stronger or more recent baselines.
>
> **Response**
>
> We appreciate this concern. We address it in four parts.
>
> ### part 1. Scope Constraint: Interaction-Only Methods
>
> Our setting assumes only user-item interaction sequences are available: no item features, user profiles, knowledge graphs, or review text. This rules out many graph-augmented, content-based, or side-information-heavy baselines. We state this scope constraint explicitly in the revised paper.
>
> The remaining candidate categories are:
>
> | Category | Representative Works | Included? | Reason |
> | --- | --- | --- | --- |
> | Implicit knowledge transfer via combined-data training | SASRec, LinRec | Yes | Strong interaction-only baselines that learn from combined source/target interactions without explicit alignment |
> | Source-assisted transfer (no cross-domain overlap) | MetaTL, TL | Yes (both) | Core comparison |
> | Contrastive self-supervised (sequential) | CL4SRec, DuoRec, CLCR | **New: CL4SRec** | Most-cited contrastive sequential method operating purely on interaction sequences. CLCR requires additional side information.|
> | Cross-domain sequential (user-overlap based) | AMID, CDRIB | **New: AMID** | Most recent open-world cross-domain method (partial user overlap, WWW 2024). Added in revision. Not strict requirement for user overlapping between source and target. Explictly use the similarity for knowledge transfer. |
> | Graph / content-based cross-domain | MIFN, BiTGCF | Not applicable | Require side features/relation graph |
>
> **Why CL4SRec and AMID.** We add one representative from each applicable category missing from the original submission.
>
> - **CL4SRec** is chosen over DuoRec because DuoRec's supervised view relies on finding semantically similar sequences, which is unreliable when target users have only 4 interactions. CL4SRec's data-level augmentations (crop, mask, reorder) are agnostic to sequence length and therefore give the contrastive paradigm its best chance under our cold-start setting.
>
> - **AMID** is chosen because it is the most recent cross-domain sequential method (WWW 2024) designed under **open-world assumptions** — it has no strict requirements on user overlap between domains and explicitly accommodates non-overlapping users via its InnerComp/InterComp interest-group mechanism. This is the close available relaxation toward our setting, where source and target populations are disjoint. Cross-domain sequential recommendation traditionally relies on shared/overlapping **users** as the bridge. For example, CDRIB assumes *full* user overlap between the two domains (the shared users serve as the cross-domain bridge via variational information bottleneck), which does not hold in our cold-start formulation.
>
> ### References for Baselines
>
> - **SASRec** — Wang-Cheng Kang and Julian McAuley. "Self-Attentive Sequential Recommendation." ICDM 2018.
> - **BERT4Rec** — Fei Sun, et al.. "BERT4Rec: Sequential Recommendation with Bidirectional Encoder Representations from Transformer." CIKM 2019.
> - **MetaTL** — Jianling Wang, et al.. "Sequential Recommendation for Cold-start Users with Meta Transitional Learning." SIGIR 2021.
> - **CL4SRec** — Xu Xie, et al. "Contrastive Learning for Sequential Recommendation." ICDE 2022.
> - **DuoRec** — Ruihong Qiu, et al. "Contrastive Learning for Representation Degeneration Problem in Sequential Recommendation." WSDM 2022.
> - **AMID** — Wujiang Xu, et al. "Rethinking Cross-Domain Sequential Recommendation under Open-World Assumptions." WWW 2024.
> - **CDRIB** — Jiangxia Cao, et al. "Cross-Domain Recommendation to Cold-Start Users via Variational Information Bottleneck." ICDE 2022.
> - **BiTGCF** — Meng Liu, et al. "Cross Domain Recommendation via Bi-directional Transfer Graph Collaborative Filtering Networks." CIKM 2020.
> - **LinRec** — Langming Liu, et al. "LinRec: Linear Attention Mechanism for Long-term Sequential Recommender Systems." SIGIR 2023.
> - **SIGMA** — Ziwei Liu, et al. "SIGMA: Selective Gated Mamba for Sequential Recommendation." AAAI 2025.
> - **MIFN** — Muyang Ma, et al. "Exploring Mixed Information Flow for Cross-domain Sequential Recommendations." ACM Trans. Knowl. Discov. Data 16(4), 2022.
> - **CLCR** — Yinwei Wei, et al. "Contrastive Learning for Cold-Start Recommendation." ACM Multimedia 2021.

---

> ### Author Response · Authors · 2026-04-21
> **Response to the Baseline issue from Reviewer 4dmv**
>
> ### part 2. New Baseline: CL4SRec (Contrastive Sequential)
>
> **CL4SRec** applies contrastive augmentation (crop / mask / reorder) on all available interaction data to improve sequence representation. However, without explicit separation between source and cold-start populations, a natural question arises: can joint contrastive learning over combined users effectively transfer to cold-start users, or does it potentially conflate different preference regimes?
>
> Our method instead partitions users into source (long-history) and cold-start (short-history) groups and transfers through an intermediate simulated target domain.
>
> We tune CL4SRec in two stages: first optimizing the backbone without the contrastive loss, and then jointly tuning the backbone with CL-specific parameters (cl_weight, temperature, cl_similarity). This strategy is more efficient and yields better performance.
>
> #### CL4SRec vs. DACSR++ — Time-Split, max\_target\_len = 4, Sampled Evaluation (NDCG@10)
>
> | Dataset | SASRec | CL4SRec | DACSR++ | Impr. (DACSR++ vs CL4SRec) |
> | --- | ---: | ---: | ---: | ---: |
> | Beauty | 0.5687 | 0.5668 | **0.5967** | +5.28% |
> | Sports | 0.4859 | 0.4985 | **0.5710** | +14.54% |
> | Toy | 0.4967 | 0.4632 | **0.5763** | +24.41% |
> | Kindle | 0.8660 | 0.9052 | **0.9320** | +2.96% |
> | Electronics | 0.4568 | 0.4337 | **0.4854** | +11.92% |
>
> > **Note:** Compared to the SASRec backbone without contrastive learning, CL4SRec does not consistently achieve better performance under timestamp split.
>
> #### CL4SRec vs. DACSR++ — Random-Split, max\_target\_len = 4, Sampled Evaluation (NDCG@10)
>
> | Dataset | SASRec | CL4SRec | DACSR++ | Impr. (DACSR++ vs CL4SRec) |
> | --- | ---: | ---: | ---: | ---: |
> | Beauty | 0.3450 | 0.3465 | **0.4357** | +25.74% |
> | Sports | 0.3911 | 0.4000 | **0.4795** | +20.02% |
> | Toy | 0.3294 | 0.3304 | **0.4510** | +36.5% |
> | Kindle | 0.5430 | 0.5510 | **0.5695** | +3.36% |
> | Electronics | 0.3871 | 0.3861 | **0.4433** | +14.82% |
>
> > **Note:**  CL4SRec basically achieves better or close performance than its SASRec backbone under random split.
>
> #### CL4SRec vs. All Baselines on ML-1M, max\_target\_len = 4, Full-Item Ranking (NDCG@10) (newly added data, the detailed data info will be provided in another response)
>
> | Model | NDCG@10 | Recall@10 | MRR |
> | --- | ---: | ---: | ---: |
> | SIGMA | 0.0792 | 0.1198 | 0.0783 |
> | SASRec | 0.0917 | 0.1451 | 0.0880 |
> | LinRec | 0.1050 | 0.1750 | 0.0964 |
> | CL4SRec | 0.1086 | 0.1748 | 0.1001 |
> | **DACSR++** | **0.1129** | 0.1862 | **0.1036** |
> | TL | 0.1112 | **0.1892** | 0.0999 |
>
> On ML-1M, CL4SRec  sits below DACSR++  and just above LinRec. DACSR++ is strongest on NDCG@10 and MRR, while TL remains strongest on Recall@10. However, the ML-1M by\_length split has substantial temporal overlap between source and target users (see details in the response to new dataset), making it closer to a random-holdout setting than a strict cold-start scenario.
>
> Under the stricter time-based splits on Amazon datasets, the comparative results are clearer.  This pattern is consistent with our interpretation: CL4SRec improves general sequence representation, but without DACSR++'s explicit source-to-cold-start transfer structure it remains weaker under strict cold-start conditions.
>
> The full results of new experiments on CL4SRec with other metrics will be added in the revised paper.
>
> **Tuning methodology note:** To ensure fair comparison, CL4SRec was tuned in two stages: (1) first optimizing the backbone parameters without the contrastive loss, using the same training schedule and hyperparameters as DACSR++, and (2) then jointly tuning the backbone and CL-specific parameters (`cl_weight`, `temperature`, `cl_similarity`) on the validation set. This two-stage approach prevents the contrastive loss from dominating early training and allows the base encoder to learn meaningful representations before contrastive weighting is applied.

---

> ### Author Response · Authors · 2026-04-21
> **Response to the Baseline issue from Reviewer 4dmv**
>
> ### part 3. New Baseline: AMID (Cross-Domain model)
>
> **AMID** is a cross-domain sequential method designed for **open-world** assumptions: unlike CDSR methods that require fully overlapping users across domains, AMID accommodates *partial* user overlap by constructing user interest groups via batch-level InnerComp/InterComp comparisons that include both overlapping and non-overlapping users. It is the most recent cross-domain sequential baseline that relaxes the shared-user bridge assumption, making it the closest available fit to our disjoint source/target cold-start setting.
>
> #### AMID Results on ML-1M, max\_target\_len = 4, Full-Item Ranking
>
> | Model | NDCG@10 | Recall@10 | MRR |
> | --- | ---: | ---: | ---: |
> | AMID | 0.0552 | 0.1105 | 0.0506 |
> | SASRec | 0.0917 | 0.1451 | 0.0880 |
> | **DACSR++** | **0.1129** | 0.1862 | **0.1036** |
>
> #### AMID Results on Amazon Datasets, max\_target\_len = 4, Sampled Evaluation (100 neg, NDCG@10)
>
> | Dataset | DACSR++ (random) | AMID (random) | DACSR++ (timesplit) | AMID (timesplit) |
> | --- | ---: | ---: | ---: | ---: |
> | Beauty | **0.4357** | 0.1095 | **0.5967** | 0.2765 |
> | Sports | **0.4795** | 0.1239 | **0.5710** | 0.2115 |
> | Toy | **0.4510** | 0.0705 | **0.5763** | 0.1814 |
> | Kindle | **0.5758** | 0.0662 | **0.9320** | 0.2207 |
> | Electronics | **0.4433** | 0.1611 | **0.4854** | 0.3182 |
>
> AMID performs substantially worse than DACSR++ across all five datasets under both splits. A likely reason is its cross-domain interest grouping mechanism, which clusters users from both source and target domains and assumes that target representations are sufficiently reliable. With only four interactions per target user, these representations are too noisy for stable similarity estimation. Although AMID is the closest method to a domain-adaptation approach, its reliance on target-side similarity introduces noise and degrades performance in this setting.
>
> The full results of AMID will be added into the revised manuscript.
>
> ### part 4. Why DACSR++ Has a Structural Advantage Over Both New Baselines
>
> The results above follow from a structural difference in how each method uses the source domain.
>
> **vs. CL4SRec:** CL4SRec applies contrastive augmentation (crop/mask/reorder) uniformly across all users without distinguishing source from cold-start users. Its positive pairs are augmented views of the same sequence — there is no mechanism to align representations across the long/short boundary between source and target populations, and no use of inter-user similarity structure from the source. DACSR++ instead uses a simulated target domain (short subsequences of source users) to impose two simultaneous constraints: (1) the same user's long and short sequences should produce similar representations, and (2) users that are similar in the source domain should remain similar in the short-sequence regime. This trains the encoder to be structurally robust to the cold-start length gap.
>
> **vs. AMID:** AMID computes user interest groups via batch-level cross-domain similarity (InnerComp/InterComp), which is conceptually close to DACSR++'s similarity-guided transfer. The critical difference is the direction of information flow. AMID treats the similarity computation as symmetric — it groups users from both source and target sides together, relying on the target representations being reliable enough to form coherent clusters. With only 4 target interactions, those representations are too noisy for stable grouping. DACSR++ avoids this by computing similarity exclusively from the source side (long, reliable histories) and imposing it as a unidirectional training signal onto the simulated short-sequence space. The information flows source→ simulated target→target, not as a symmetric exchange.
>
> This distinction explains the pattern across both baselines: when the challenge is mainly sparsity (ML-1M `by_length`), CL4SRec partially closes the gap; when genuine temporal shift is present (Amazon `by_timestamp`), the structured source→target design of DACSR++ matters more. AMID remains weak because the 4-item target side is too noisy for stable symmetric grouping.
>
> **Note:** We use the default AMID implementation from the authors’ official repository, including full InnerComp/InterComp similarity, adapted to our source–target setting. We do not perform extensive hyperparameter tuning due to its consistently poor performance.

---

> ### Author Response · Authors · 2026-04-21
> **Response to Reviewer 4dmv on the Dataset Diversity**
>
> **Reviewer concern.**
> Primary datasets come from Amazon sharing similar behavior, results may not generalize. Whether at least one non-Amazon dataset can be included, eg., MovieLens, Yelp, and industrial logs.
>
> **Response:**
>
> We agree that cross-platform diversity strengthens generalization claims. We add **MovieLens-1M (ML-1M)**  and report results for all methods at `max_target_len = 4`.
>
> ### ML-1M:
>
> MovieLens-1M is a classic movie rating dataset from the GroupLens research lab, collected from the MovieLens website. Its properties differ from the Amazon datasets with much denser user item sequences, which provides a complementary evaluation axis.
>
> ### Split Design for a Surrogate Cold-Start
>
> ML-1M does not yield a meaningful strict temporal cold-start split because it is a static snapshot  rather than a live platform log:
>
> **Observation 1: Almost no new users appear after any reasonable timestamp cutoff.**
>
> The table below show users qualify as "new" (first interaction on or after the cutoff) under a by_timestamp split at various cutoffs (total users = 6,038):
>
> | Cutoff date | New users after cutoff | % of all users |
> | --- | ---: | ---: |
> | Dec 2000  | 343 | 5.7% |
> | Jun 2001  | 6 | 0.1% |
> | Jan 2002  | 3 | 0.0% |
> | Jul 2002  | 2 | 0.0% |
>
> Almost all ML-1M users first appear in the early months of the dataset
>
> **Observation 2: The "new" users from the earliest feasible cutoff are not cold-start users.**
>
> The most cutoff (Dec 2000, 343 users) does not produce cold-start users with the interaction-length breakdown below:
>
> | Interaction-length bin | # users | Notes |
> | --- | ---: | --- |
> | 1–4| **0** | removed at preprocessing |
> | 5–9 | 8 | Borderline sparse |
> | 10–19 | 68 | Medium history |
> | 20+ | 267 | Long-history users |
>
> The 343 "new" users have an average of 70.5 interactions — they are in fact long-history users who happen to have registered slightly later.
>
> Instead, we adopt a **`by_length` split**: users with 4–20 total interactions form the target set and users with more than 20 interactions form the source set. This operationalizes cold-start as *short interaction history* rather than *temporal novelty*.
>
> **Data statistics after split:**
>
> | Statistic | Value |
> | --- | --- |
> | Source users | 5,084 |
> | Target users (total) | 951 |
> | Item size | 3,525 |
> | Source seq. length (mean / median) | 110.3 / 74 |
> | Target seq. length (mean / median) | 15.1 / 16 |
>
> ### `by_length` similar to Our `by_random` Split
>
> A key property of the `by_length` split is that it does **not** create a temporal change. Source and target users were active during the same period:
>
> | Group | First interaction | Last interaction |
> | --- | --- | --- |
> | Source users | 2000-04-26 | 2003-02-28 |
> | Target users | 2000-04-26 | 2003-01-26 |
>
> An Overlapping period analysis figure (will be added in the revised manuscript) shows the first- and last-interaction timestamp distributions of source and target users are nearly indistinguishable: both the first-interaction and last-interaction distributions of source and target users concentrate in the same early months of the dataset. There is no visible temporal separation between groups. This means the ML-1M `by_length` split is **structurally similar to our `by_random` split** on Amazon: those users are sparse users from the same period, not genuinely new users from a future window. We will state explicitly that ML-1M `by_length` is a **controlled short-history holdout** that complements the stricter time-based Amazon evidence.
>
> ### All Methods, ML-1M, `max_target_len = 4`, Full-Item Ranking
>
> NDCG@10 and Recall@10 are reported.
>
> | Model | NDCG@10 | Recall@10 | MRR |
> | --- | ---: | ---: | ---: |
> | AMID | 0.0552 | 0.1105 | 0.0506 |
> | SIGMA | 0.0792 | 0.1198 | 0.0783 |
> | SASRec | 0.0917 | 0.1451 | 0.0880 |
> | LinRec | 0.1050 | 0.1750 | 0.0964 |
> | CL4SRec | 0.1086 | 0.1748 | 0.1001 |
> | **DACSR++** | **0.1129** | 0.1862 | **0.1036** |
> | TL | 0.1112 | **0.1892** | 0.0999 |
>
> **Key observations:**
>
> 1. DACSR++ is strongest on NDCG@10 and MRR, while TL is strongest on Recall@10.
> 2. CL4SRec outperforms the SASRec and LinRec on this setting, shows that contrastive augmentation can partially substitute for explicit transfer.
> 3. SIGMA is the weakest, consistent with its Amazon results.
> 4. The close contest between DACSR++, TL and CL4SRec at the top reflects the lower difficulty of the `by_length` setting — all methods benefit from the overlapping time distribution.
> 5. **AMID** (fails by a wide margin). In our setting, with only 4-item target sequences, AMID's batch-level inter/intra-domain contrastive objectives produce noisy target-side representations, making stable cross-domain cluster alignment infeasible which might be the reason AMID fails for cold-start.
>
>
> The results will be added in the revision with a note clarifying that ML-1M is a complementary benchmark to the Amazon, which is more related to controlled random split, instead of the time-based split.

---

> ### Author Response · Authors · 2026-04-21
> **Response to Reviewer 4dmv on "evaluation protocol"**
>
> **Reviewer concern.**
> The paper constructs the cold-start setting partly via an "emulated target domain" obtained by sampling short subsequences from long user histories. The reviewer questions whether this setup captures the challenges of real-world cold-start recommendation beyond sequence-length sparsity, noting that genuinely new users may exhibit different preference distributions.
>
> **Response draft.**
>
> We thank the reviewer for raising this important point. Our response has three parts: (i) clarification of the emulated target, (ii) its role as an inductive bias, and (iii) its connection to real cold-start evaluation.
>
> ### Clarification: The Emulated Target Is a Training Bridge, Not the Evaluation Target
>
> We would like to clarify again that the evaluation target domain under the `by_timestamp` split, are these users who appear only after a hard temporal cutoff T and are never seen in the source. Their test interactions are drawn from a future window the source model has never observed.
>
> The **emulated target** — short subsequences sampled from source users' histories — is used only during **training**, as an intermediate bridge with purposes:
>
> | Purpose | Description |
> | --- | --- |
> | **Short-sequence robustness** | Exposing the model to truncated interaction sequences so it can handle more easily the short histories of real cold-start users at inference time. |
> | **Similarity transfer signal** | Providing a proxy for cold-start users so that the similarity-based alignment objective can be computed during training, before any real target users are seen. |
> | **Popularity-debiased pooling** | Enabling inverse-popularity weighting in the simulated short-sequence representation, so the training signal is not dominated by a few highly frequent items and instead emphasizes the more informative preference structure that should transfer to real cold-start users. |
>
> The emulated auxiliary target is discarded at evaluation. The reported metrics are computed entirely on the held-out real cold-start users — those who genuinely appear after the temporal cutoff.
>
> ### Why the Simulated Target Is the Right Inductive Bias
>
> Base on the purposes above, the simulated target is the mechanism that gives the contrastive objective the right structure.
>
> 1. **Length-invariant user identity**: the same source user's full-length sequence and their simulated short-length subsequence should produce similar representations. This directly trains the encoder to be robust to the length difference between source and cold-start users — something similar to sequence-level augmentation (CL4SRec's crop/mask/reorder), but explicitly transferred to the target model representing cold-start users.
>
> 2. **Cross-user similarity transfer**: Cross-user similarity transfer. Users identified as similar in the source domain (via long histories) are encouraged to remain close in the simulated short-sequence space, enforcing consistency of preference structure under sparsity and generalizing to real cold-start users at inference. AMID is conceptually related but relies on batch-level cross-domain grouping (InnerComp/InterComp), which depends on reliable target representations. With only a few interactions, target embeddings are noisy, leading to unstable grouping. In contrast, DACSR++ computes similarity solely from the source side and transfers it to short sequences (source→target), avoiding this instability.
>
> ### Connects to Real Cold-Start
>
> The key point is that whether our `by_timestamp` setting is close to the real cold-start setting. Under this split, target users are genuinely new users who appear after the temporal cutoff, so the task includes both **short target history** and **temporal / item-distribution shift** relative to the source side. This defines a realistic and challenging cold-start regime.
>
> The emulated target plays a different role: it is a training bridge, not a proxy for evaluation. The sampled short subsequences from long-history source users carry source-side information in the short-sequence regime, like inherit reliable identity and similarity structure from the source users, while matching the length constraint.
>
> In summary, the emulated target addresses the length mismatch during training, while the by_timestamp split evaluates transfer to real new users under both sparsity and distribution shift.

---

### Review · Reviewer_NmQF · 2026-04-23

**Summary Of Contributions:**

Summary:

The paper investigates the cold-start problem in sequential recommendation by reformulating it as a domain adaptation challenge. The authors define regular users with rich interaction histories as the source domain and cold-start users as the target domain. To bridge the representation gap between these domains, the authors propose the DACSR (Domain Adaptation for Cold-Start Recommendation) framework. The architecture utilizes a dual-transformer network with shared item embeddings. To facilitate knowledge transfer, the framework generates an emulated target domain by randomly subsampling sequences from the source domain, and then aligns the contextual representations using contrastive learning. The authors progressively enhance the model to handle complex shifts by introducing popularity-adjusted user representations (DACSR+) and a similarity-weighted contrastive loss (DACSR++).

Strengths:

S1: Conceptualizing the cold-start problem through the lens of domain adaptation by specifically decomposing it into length shift and complex shift is an interesting approach.

S2: The step-by-step enhancement from DACSR to DACSR++ is logically sound, and the ablation studies clearly demonstrate the efficacy of each added component.

S3: The experimental validation across five public Amazon datasets, utilizing both random and time-based splits, robustly demonstrates the framework's superiority over strong baselines.

Weaknesses:

W1: The assumption that randomly subsampling a regular user's history accurately reflects true cold-start behavior is a weak inductive bias. Real cold-start users often exhibit distinct exploratory behaviors (e.g., interacting strictly with globally popular items) rather than acting as a randomly missing subset of a long-term user.

W2: The inverse popularity weighting strategy introduced in DACSR+ poses a risk of numerical instability or representation collapse if exposed to extreme long-tail items.

W3: The dual-transformer architecture coupled with an $O(N^2)$ in-batch similarity-weighted contrastive loss introduces substantial memory and computational burdens, which are not thoroughly quantified in the submission.

**Audience:**

Yes

**Audience Explanation:**

The framework’s ability to handle complex shift in sequential recommendation without requiring auxiliary side-information is appealing to the TMLR audience, including researchers focused on recommender systems, representation learning, and data distribution shifts.

**Broader Impact Concerns:**

The research focuses on a fundamental algorithmic improvement for sequential recommendation systems. There are no immediate or direct negative ethical implications introduced by this specific methodology. However, as with all highly effective recommender systems, improving cold-start accuracy means new users may be profiled and placed into behavioral echo chambers more rapidly than in legacy systems. A brief sentence acknowledging standard recommender system biases in the broader impact or conclusion section would suffice.

**Claims And Evidence:**

Yes

**Claims Explanation:**

The empirical results support the core claims.

**Requested Changes:**

Under the time-based split, randomly sampling the source domain risks negative transfer by forcing the model to learn outdated concepts that target users no longer interact with. The paper would be strengthened by discussing a target-aware subsampling strategy. Drawing inspiration from data selection methods used in concept drift (distribution shift) literature [1], the model could selectively sample that exhibit distributional similarity to the current target trends, rather than utilizing uniform random probability.

[1] Quilt: Robust Data Segment Selection against Concept Drifts

---

> ### Author Response · Authors · 2026-05-03
> **Response to comment "W3" from Reviewer NmQF**
>
> We thank the reviewer for raising this important concern. We clarify that the computational and memory overhead can be more precisely characterized as follows theoretically and empirically.
>
> **1. The O(N²) similarity is over batch size**
>
> The pairwise similarity matrix is computed within a single training batch of emulated target users. For a batch of B users with d-dimensional embeddings, this requires B² dot products each of dimension d, costing O(B²·d) per batch. With N total training users, each epoch runs N/B batches, giving a total per-epoch similarity cost of O(N·B·d), which scales linearly in N (the number of users).
>
> **2. The dual-encoder overhead is bounded and partially offset by shared item embeddings.**
>
> The source and target encoders share item embeddings, which are typically the largest parameter block in a sequential recommendation model (vocabulary size × embedding dimension). The additional parameters relative to a single SASRec model are the transformer layer weights of the second (target) encoder. The shared item embedding table reduces the memory cost compared to two fully independent models.
>
> **Concrete complexity comparison.** We use the following notation throughout: $V$ = vocabulary size, $d$ = hidden dimension, $n_b$ = number of transformer blocks, $L_s$ = source sequence length, $L_t$ = target (cold-start) sequence length, $B$ = batch size, $N$ = number of training users.
>
> Following standard Transformer complexity analysis, the per-block cost for a sequence of length (L) is (O(L d^2)) for linear projections and feed-forward layers, and (O(L^2 d)) for self-attention. The in-batch similarity matrix adds $O(B^2 \cdot d)$ per batch ($B^2$ dot products each of dimension $d$), giving a total per-epoch similarity cost of $O(N \cdot B \cdot d)$ — linear in $N$.
>
>
> *Model parameters thus:*
>
> | Component | SASRec | DACSR++ |
> | --- | --- | --- |
> | Item embedding | $V \cdot d$ | $V \cdot d$ (shared) |
> | Transformer weights | $O(n_b \cdot d^2)$ | $O(2 \cdot n_b \cdot d^2)$ |
> | **Total** | $O(V \cdot d + n_b \cdot d^2)$ | $O(V \cdot d + 2 \cdot n_b \cdot d^2)$ |
>
> The item embedding table (typically the largest block when $V \gg n_b \cdot d$) is shared between both encoders, so DACSR++ adds only one extra transformer stack relative to a single SASRec model.
>
> *Computation per training step (batch of $B$ users):*
>
> | Cost term | Complexity |
> | --- | --- |
> | Source encoder | $O(B \cdot n_b \cdot (L_s \cdot d^2 + L_s^2 \cdot d))$ |
> | Target encoder on pseudo-target  | $O(B \cdot n_b \cdot (L_t \cdot d^2 + L_t^2 \cdot d))$ |
> | Target encoder on real cold-start users | $O(B \cdot n_b \cdot (L_t \cdot d^2 + L_t^2 \cdot d))$ |
> | In-batch similarity matrix | $O(B^2 \cdot d)$ |
>
> *Empirical measurements (Toy dataset for example):*
>
> | | SASRec | DACSR++ |
> | --- | ---: | ---: |
> | Active parameters | 1,256,832 | 1,356,928 |
> | **Parameter overhead** | — | **+8%** |
> | **Total per epoch** | **0.25 s** | **0.29 s** |
> | **Epoch overhead** | — | **+16%** |
>
>
> **Inference cost.** At inference time, only the target encoder is used to produce recommendations for cold-start users. The source encoder is not invoked during inference, so inference computational cost are identical to a single SASRec model.
>
> In summary, the additional training cost relative to SASRec is **+16% per epoch** empirically, arising from (i) extra encoder  and (ii) the in-batch similarity matrix. Inference cost is unchanged.
>
> **Hyperparameter search cost.** We acknowledge that DACSR++ introduces additional loss hyperparameters (mainly regularizations for the different loss terms and mask probability) compared to SASRec. When the full hyperparameter search is included, the total wall-clock cost is substantially higher than a single SASRec run.

---

> ### Author Response · Authors · 2026-05-03
> **Response to comment "W2" from Reviewer NmQF**
>
> **Reviewer concern**
> The inverse popularity weighting strategy introduced in DACSR+ poses a risk of numerical instability or representation collapse if exposed to extreme long-tail items.
>
> **Response**
>
> We thank the reviewer for this concern. The inverse popularity weighting is **bounded by construction**. Item popularity is normalised to $[0, 1]$, and the weight function is:
>
> $(w(P) = 0.1 / P )  if  ( P \ge 0.1 ); otherwise  ( w(P) = 1 )$
>
> This guarantees all weights lie in $[0.1, 1]$ with a maximum amplification ratio of **10×** (rare items vs. popular items), preventing arbitrarily large weights or gradients. We acknowledge this design detail was not sufficiently described in the manuscript and will include it in the revision.
>
> **On representation collapse:** the weighting acts as a bounded rebalancing mechanism that increases the influence of less popular items without suppressing frequent ones. The representation remains a normalized mixture over all items in the sequence, preserving diversity rather than collapsing toward rare-item embeddings.
>
> **Empirically**, we observe stable convergence across all five Amazon datasets, with no gradient explosion or training instability. We will include training loss curves across all five datasets as a supplementary figure in the revision, demonstrating smooth decrease in training loss  and increasing in validation performance throughout training.

---

> ### Author Response · Authors · 2026-05-03
> **Response to comment "W1" from Reviewer NmQF**
>
> **Reviewer concern:**
> The assumption that randomly subsampling a regular user's history accurately reflects true cold-start behavior is a weak inductive bias.
>
> **Response:**
>
> We agree that random subsampling is a simplified approximation and does not fully capture real cold-start behavior. Our design mitigates this mismatch in two ways.
>
> **(1) Supervision from real cold-start data.**
> The target encoder is trained not only on emulated short sequences but also on **real cold-start users** in the training split. As a result, genuine cold-start behaviors are directly learned by the encoder. The emulated sequences serve as an auxiliary signal rather than the sole source of supervision, so the distributional gap does not create a critical blind spot.
>
> **(2) Identity-based similarity.**
> Our inter-user similarity is defined at the **user level** (long sequence vs. its own short subsequence), encouraging the model to preserve user identity under truncation and to map short sequences into the source representation space. This objective does not rely on the emulated sequence matching the exact item distribution of real cold-start users; the supervisory signal flows through the identity.
>
> We view random subsampling as a **neutral truncation mechanism** to learn robustness to limited histories. We agree that incorporating **target-aware** could better reflect exploratory behavior. Empirically, we observe consistent gains over baselines across datasets, suggesting that the method is robust to this approximation.

---

> ### Author Response · Authors · 2026-05-03
> **Response to comment "Negative Transfer from Outdated Source Concepts" from Reviewer NmQF**
>
> **Reviewer concern.**
> Negative Transfer from Outdated Source Concepts under the time-based split by randomly sampling the source domain. The paper would be strengthened by discussing a target-aware subsampling strategy. Drawing inspiration from "quilt", the model could sample source users that exhibit distributional similarity to current target.
>
> **Response:**
>
> We agree that temporal drift can cause negative transfer and that target-aware source sampling is a natural extension. We respond at two levels.
>
> **Existing mitigating mechanisms.** Two strategies limit the effect of old source concepts: (1) the target encoder is trained  on real cold-start users, so post-cutoff behaviors have direct gradient access; (2) the contrastive loss weight  is tuned on the cold-start validation split. If old source similarity were harmful, validation would penalize it.
>
> **QUILT-inspired temporal source sampling.** We borrow two high-level ideas from QUILT: temporal segmentation and Bayesian optimisation over the segment policy. Our adaptation, however, differs from the original QUILT in the following ways.
>
> QUILT partitions the data into temporal windows and may discard old windows entirely, retaining only the segments deemed relevant. When we attempted an analogous strategy, restricting training to a subset of source users,  model performance dropped substantially, because removing source users reduces the effective training. We therefore preserve *all* source users in training and instead vary how each user's history is sub-sampled.
>
> Specifically, source users are ranked by their last interaction timestamp and divided into $K$ temporal segments in our adaptation, and $p_k$ is the per-item keep probability when constructing the simulated target sequence for source users in segment $k$.  For a user in segment $k$, each item is kept independently with probability $p_k$, producing a simulated cold-start sequence of variable length. Theoretically, less relevant segment users can receive a lower $p_k$, making their simulated target sequences shorter and less representative of active engagement, while relevant-segment users receive a higher $p_k$. The vector $[p_0,\ldots,p_{K-1}]$ with $K \in \{1,2,3,5,7\}$ and $\beta_{SS}$ for the supervision of emulated target loss are jointly tuned by Bayesian optimisation on cold-start validation NDCG@10, with uniform $p_k = p$, the optimal sampling rate of DACSR++, as the warm-start probe .
>
> **Empirical results.** We evaluated across five Amazon datasets under both cold-start splits (NDCG@10):
>
> *By-timestamp split:*
>
> | Dataset | DACSR++ *p* | DACSR++ NDCG@10 | QUILT *K* | QUILT *p_k* | QUILT NDCG@10 | Δ (%) |
> |---|---:|---:|---:|---|---:|---:|
> | Beauty | 0.60 | 0.5967 | 5 | [0.49, 0.22, 0.80, 0.65, 0.63] | 0.6005 | +0.64 |
> | Toy | 0.60 | 0.5763 | 2 | [0.60, 0.38] | 0.5806 | +0.75 |
> | Sport | 0.60 | 0.5710 | 2 | [0.78, 0.75] | 0.5672 | −0.67 |
> | Kindle | 0.70 | 0.9191 | 2 | [0.48, 0.34] | 0.9290 | +1.08 |
> | Electronics | 0.60 | 0.4854 | 3 | [0.67, 0.75, 0.64] | 0.4850 | −0.08 |
>
> *By-random split:*
>
> | Dataset | DACSR++ *p* | DACSR++ NDCG@10 | QUILT *K* | QUILT *p_k* | QUILT NDCG@10 | Δ (%) |
> |---|---:|---:|---:|---|---:|---:|
> | Beauty | 0.60 | 0.4357 | 3 | [0.75, 0.40, 0.42] | 0.4430 | +1.68 |
> | Toy | 0.50 | 0.4510 | 7 | [0.89, 0.81, 0.90, 0.46, 0.77, 0.31, 0.88] | 0.4657 | +3.26 |
> | Sport | 0.50 | 0.4795 | 3 | [0.26, 0.81, 0.62] | 0.4881 | +1.79 |
> | Kindle | 0.40 | 0.5540 | 1 | [0.54] | 0.5605 | +1.17 |
> | Electronics | 0.40 | 0.4433 | 3 | [0.46, 0.87, 0.71] | 0.4566 | +3.00 |
>
> Under random split, temporal segmentation consistently improves a bit the NDCG@10 across all five datasets (+1.17% to +3.26%), with  non-uniform learned $p_k$, confirming that by assigning different $p_k$ per cohort, BO can tune how aggressively each cohort's history is sub-sampled when constructing simulated cold-start sequences.  Under time-based split, gains are more mixed (three datasets improve, two show slight declines), and overall slight. Cold-start users already occupy the most recent period, so the target encoder's direct cold-start gradient path and $\beta_{SS}$ tuned on cold-start validation already attenuate the influence of old source concepts. Temporal re-weighting of source users provides limited additional headroom over these existing mechanisms. We will include the complete results in the revision.
>
> **Tuning cost of the QUILT adaptation.** The QUILT adaptation adds $K$, $p_k$, and $\beta_{SS}$ on top of the existing DACSR++ hyperparameters, tuned jointly via Bayesian optimisation. BO does not guarantee a global optimum but is empirically effective without manual grid search. We acknowledge the additional tuning cost and will detail the procedure in the supplementary material.

---

### Comment · Action_Editor_PeqV · 2026-04-28
**Response to reviews**

Dear authors:

The author response phase has started, please note that you have up to two weeks from the start of the discussion phase to post your author response. We encourage you to address all reviewer comments and questions as clearly and concretely as possible.

Best,
AC

---

> ### Author Response · Authors · 2026-05-05
> **Summary of the changes made according to the reviewers' comments**
>
> We have revised the paper in the following key aspects. For ease of reference, the main revisions are highlighted in red.
>
> First, we expanded the related work to better position our problem and method in Section 2. We added a discussion of cross-domain sequential recommendation (CDSR), including EMCDR, CDRIB, AMID, HeroGraph, MIFN, and BiTGCF. We clarify that most CDSR methods rely on shared users, item-side knowledge, or auxiliary content, while our setting assumes disjoint source and target user populations. We also added a dedicated positioning paragraph explaining how our DACSR differs from implicit combined training, transfer learning, meta-learning, contrastive sequential recommendation, and open-world CDSR. References are updated accordingly.
>
> Second, we clarified the role of the emulated target domain in Section 3.1. The revised text explains that it serves two purposes: length-invariant user alignment between each user's full source sequence and simulated short sequence, and cross-user similarity transfer from reliable long-history source users to the short-sequence regime.
>
> Third, we added a discussion of alternative target-aware sampling in Section 3.3. While the main method uses uniform random subsampling, we mention a QUILT-inspired temporally stratified sampling extension.
>
> Fourth, we refined the popularity-weighted user representation in Section 3.4. We now explicitly define the bounded inverse-popularity weight, explain its range and maximum amplification ratio, and clarify that it rebalances frequent and rare items without allowing unstable weights or collapsing representations toward rare items.
>
> Fifth, we strengthened the experimental setup and baselines in Section 4.1 and 4.2. We clarify that the random split should be interpreted as a controlled length-reduction experiment rather than a realistic temporal cold-start scenario. We also add CL4SRec as a contrastive-learning baseline and AMID as an open-world CDSR/domain-adaptation baseline that does not require fully shared users.
>
> Sixth, we updated the main performance tables and captions in Table 2 and 3. We report CL4SRec and AMID results, clarify how the improvement is computed, and explain the tuning protocol for CL4SRec and implementation choice for AMID. We also added analysis explaining why CL4SRec helps in the random split but is less consistent under the time-based split, and why AMID performs poorly when target users have only a few interactions in Section 4.3.
>
> Seventh, we added full-item ranking evaluation under the time-based split in Table 13 and 14 in Appendix E. These supplementary results show that the conclusions are consistent with that of the 100-negative evaluation protocol.
>
> Eighth, we expanded the ablation and model-variant discussion in Section 4.4. We clarify that DACSR++ is generally strongest, but Kindle is an exception likely due to weaker source-domain user similarity. We support this with a user-similarity analysis in the appendix G. Table 6 is updated to include more baseline models.
>
> Ninth, we added a computational complexity in Section 5. This section analyzes the cost of in-batch similarity computation, the dual-encoder parameter overhead, training-time complexity, empirical runtime overhead, inference cost, and additional hyperparameter tuning cost.
>
> Tenth, we added complementary experiments on ML-1M in Appendix H. We explain why ML-1M does not support a meaningful strict timestamp-based cold-start split, justify the use of a length-based split, and report full-item ranking results as a complementary short-history benchmark.
>
> Finally, we added a QUILT-inspired target-aware sampling extension in the Appendix I. This section describes how temporal segments are assigned different sampling probabilities, reports results under both random and time-based splits, and discusses the additional Bayesian-optimization tuning cost.
>
> This revised version will be uploaded to replace the original one.

---

### Decision · Action_Editor_PeqV · 2026-05-31

**Recommendation:** Accept with minor revision

**Additional Comments:**

The reviewers generally provide positive final ratings. There are some critical comments about the newly added experiments, please address these concerns in the final version.

"The newly added experiments some important details remain unclear, including tuning fairness, implementation reliability of the new baselines, and how much the results depend on the specific split construction. The new MovieLens-1M result is useful but is acknowledged by the authors to be closer to a controlled short-history setting than a strict temporal cold-start setting. The main evidence therefore still relies heavily on Amazon-derived datasets."

**Audience:**

Yes

**Audience Explanation:**

Researchers in the domain adaptation/transfer learning field will be interested in this work.

**Claims And Evidence:**

Yes

**Claims Explanation:**

This paper studies cold-start sequential recommendation through the lens of domain adaptation, treating users with long interaction histories as the source domain and cold-start users as the target domain. The proposed DACSR framework uses a dual-transformer architecture with shared item embeddings, constructs an emulated target domain by subsampling source sequences, and aligns representations via contrastive learning. The paper further introduces DACSR+ and DACSR++ to better account for popularity bias and user similarity. Reviewers found the problem formulation interesting and the empirical results on generally strong, with consistent improvements over baselines. There are some concerns about positioning to the existing literature, justification of the formulation, evaluation and dataset diversity, and have been addressed in the response. Overall, the paper offers an interesting perspective on cold-start recommendation with encouraging results.